# FAT10 and NUB1L cooperate to activate the 26S proteasome

Florian Brockmann[1], Nicola Catone[2], Christine Wünsch[1], Fabian Offensperger[3], Martin Scheffner[3] , Gunter Schmidtke[1], Annette Aichem[1,2]

**The interaction of the 19S regulatory particle of the 26S proteasome with ubiquitylated proteins leads to gate opening of the 20S core particle and increases its proteolytic activity by binding of the ubiquitin chain to the inhibitory deubiquitylation enzyme USP14 on the 19S regulatory subunit RPN1. Covalent modification of proteins with the cytokine inducible ubiquitin-like modifier FAT10 is an alternative signal for proteasomal degradation. Here, we report that FAT10 and its interaction partner NUB1L facilitate the gate opening of the 20S proteasome in an ubiquitin- and USP14-independent manner. We also show that FAT10 is capable to activate all peptidolytic activities of the 26S proteasome, however only together with NUB1L, by binding to the UBA domains of NUB1L and thereby interfering with NUB1L dimerization. The binding of FAT10 to NUB1L leads to an increased affinity of NUB1L for the subunit RPN1. In conclusion, the herein described cooperation of FAT10 and NUB1L is a substrate-induced mechanism to activate the 26S proteasome.**

## Introduction

The 26S proteasome is the central protein degradation complex in the cytoplasm and nucleus of eukaryotic cells. It consists of the 20S core particle (CP) and one or two copies of the 19S regulator (Wehmer et al, 2017). The three proteolytic subunits in the 20S core particle execute the degradation of most proteins in the cell, whereas the 19S regulator is an important factor for the selection of marked proteins and their funneling into the core particle. In general, proteins are marked for degradation by a poly-ubiquitin chain of four or more covalently linked ubiquitin (Ub) moieties, whose C-termini are linked via isopeptide bonds to lysines of another ubiquitin (Thrower et al, 2000). Except for binding ubiquitylated proteins, the 19S regulator also deubiquitylates those proteins, unfolds them, and translocates them into the 20S core particle (Bard et al, 2019). Several studies have shown that the

binding of ubiquitin conjugates can lead to a gate opening of the 20S proteasome. This happens via an ATP-dependent mechanism of the two 19S regulatory subunits RPT2 and RPT5, which bind to pockets at the outer ring of the 20S proteasome and widen the gate formed by the N-termini of the 20S $\alpha$-subunits. The poly-ubiquitin chain is rescued in this process from degradation by RPN11, a deubiquitylation enzyme (DUB), which is a 19S regulatory subunit that deubiquitylates proteins during the process of degradation (Verma et al, 2002; Worden et al, 2014). Although proteins deubiquitylated by RPN11 are still degraded, another DUB, called USP14 in humans and UBP6 in *Saccharomyces cerevisiae*, rescues poly-ubiquitylated substrate proteins from degradation. UBP6 binds via its N-terminal ubiquitin-like (UBL) domain at the "T2" site of RPN1 next to the "T1" site, where poly-ubiquitin conjugates and substrate shuttle factors, like Rad23, bind (Shi et al, 2016). USP14/UBP6 removes the ubiquitin chains en bloc from the distal end of the protein so that only one ubiquitin is left on the substrate (Lee et al, 2016). Apart from Ub removal, USP14 can also inhibit the 26S proteasome ATPase activity in a non-enzymatic way (Hanna et al, 2006; Kim & Goldberg, 2017). In contrast to these inhibitory effects, USP14 activates the gate opening of the 26S proteasome when it is bound by ubiquitin chains which induces a conformational switch in USP14 (Peth et al, 2013; Kuo & Goldberg, 2017; Hung et al, 2022). It has been shown that via this conformational switch ubiquitylated proteins elevate the peptidolytic activity of the 26S proteasome in vitro up to twofold (Bech-Otschir et al, 2009; Peth et al, 2009). More recent studies have shown that the acceleration of the peptidolytic activity of the 26S proteasome is also activated by UBL-containing proteins and shuttling factors like RAD23 and DDI1 (Collins & Goldberg, 2020). In these cases, the isolated UBL domain sufficed to activate the 26S proteasome.

Another targeting pathway leading to proteasomal degradation of proteins is the conjugation of them to the UBL modifier HLA-F adjacent transcript 10 (FAT10). Unlike ubiquitin, it consists of two UBL domains which are connected by a short and flexible linker (Aichem et al, 2018; Aichem & Groettrup, 2020). FAT10 is conjugated to hundreds of proteins by its cognate E1 enzyme UBA6, its E2 enzyme USE1, and Parkin or other E3 ligases (Chiu et al, 2007; Aichem

[1]Division of Immunology, Department of Biology, University of Konstanz, Konstanz, Germany [2]Biotechnology Institute Thurgau at the University of Konstanz, Kreuzlingen, Switzerland [3]Division of Biochemistry, Department of Biology, University of Konstanz, Konstanz, Germany

Correspondence: Annette.Aichem@bitg.ch

et al, 2010, 2012; Roverato et al, 2021). In contrast to other UBL modifiers, FAT10 targets its conjugates directly for proteasomal degradation in an ubiquitin-independent manner (Hipp et al, 2005; Schmidtke et al, 2009; Rani et al, 2012). This effect is enhanced by the expression of NEDD8 ultimate buster 1 (NUB1) and a more abundant splicing variant NUB1 long (NUB1L) (Hipp et al, 2004; Schmidtke et al, 2009). The 69.1 kD protein NUB1L consists of one UBL domain and, depending on its two splice variants, either two (NUB1) or three (NUB1L) ubiquitin-associated domains (UBA) at its C-terminus. In contrast to most UBA domains, and contrary to its name, the UBA domains of NUB1L do not bind to monomeric ubiquitin or polymeric ubiquitin chains, but to FAT10 (Raasi et al, 2001, 2005; Hipp et al, 2004). The UBL domain of NUB1L can bind to the RPN10 (hS5α) subunit of the 26S proteasome, or to RPN1 (Rani et al, 2012). In this study, we investigated whether FAT10 and NUB1L can activate the 20S gate opening and its peptidolytic activity as a potential mechanism of regulating the 26S proteasome.

# Results

### The 26S proteasome is activated by FAT10 and NUB1L together

Since the activation of the 26S proteasome by ubiquitin conjugates alone or by USP14 together with ubiquitylated proteins was previously shown (Bech-Otschir et al, 2009; Peth et al, 2009), we set out to investigate whether the ubiquitin-like modifier FAT10 can activate the 26S proteasome as well, either alone or jointly with its binding partner NUB1L. This question is pertinent as our group has shown that NUB1, and its splice variant NUB1L, bind FAT10 and enhance its degradation by the 26S proteasome in cells and in vitro (Hipp et al, 2004; Schmidtke et al, 2009). The 26S and 30S proteasomes, for simplicity referred to as 26S proteasome, were isolated with an affinity-based approach from human erythrocytes (as earlier described in the study by Besche and Goldberg [2012]). The purity and activity of the 26S proteasome was then tested on a native gel, followed by Coomassie staining and an overlay assay. In this experiment, we observed a shift upwards in molecular weight of the 26S and the 30S proteasomes after incubation with FAT10 and NUB1L (Fig 1A). In addition, the fluorescence emitted by the fluorogenic peptide Suc–LLVY–AMC after in gel digestion by the FAT10 and NUB1L bound proteasome appeared to be more intense. This change in peptide hydrolysis encouraged us to further investigate the degradation of small peptides with fluorogenic leaving groups (AMC) by the 26S proteasome. Surprisingly, the addition of FAT10 and NUB1L together but not alone stimulated the peptide hydrolysis by the 26S proteasome to a comparable extent as our positive control poly-ubiquitylated E6AP (E6AP$_{Ub}$) (Fig 1B). To ensure that the effect was due to the activation of the proteasome, and not an artifact of possible co-purified peptidases, we used the proteasome inhibitor MG132, which completely abolished Z–GGL–AMC cleavage (Fig 1C). Such an increased peptide hydrolysis was not observed with purified 20S proteasome as compared to the 26S proteasome (Fig 1D). These results led to the conclusion that the activation of the 26S proteasome by NUB1L/FAT10 relied on the 19S regulator. We were able to see an increase in fluorescence intensity when treating the purified 26S proteasome with ADP, ATP, and its

slowly hydrolysable variant ATPγS. Although ADP and ATP enabled nearly the same AMC-release shift in presence of FAT10 and NUB1L, the activation of the 26S proteasome treated with ATPγS was less prominent (Fig 1E and F) which might be due to a partial opening of the 20S gate formed by N-terminal endings of the distal 20S α-type subunit 3.

Because these findings suggested a gate opening of the 26S proteasome by NUB1L and FAT10, we performed a cycloheximide chase in WT baker's yeast and its open-gate mutant α3ΔN in which the N-termini of α3 were deleted (Choi et al, 2016) (Fig 2A). This experiment was feasible as we had shown before that ectopically expressed FAT10 was degraded in yeast and that its degradation was accelerated by non-endogenous NUB1L to a similar extent as in mammalian cells (Hipp et al, 2004; Rani et al, 2012). The degradation of transfected HA-tagged FAT10 was increased in the open-gate mutant as compared with the WT yeast. The co-expression of NUB1L in WT yeast increased the degradation of FAT10 to a level that we observed in the open-gate mutant without NUB1L, whereas NUB1L did not accelerate the degradation of FAT10 in this mutant (Fig 2A and B). This result strongly suggests that the accelerated degradation of FAT10 by NUB1L is mediated via gate opening in the 20S complex.

Investigation of the different peptidase specificities of the 20S proteasome showed the highest increase of the chymotrypsin-like activity, whereas trypsin- and caspase-like activities were only slightly less increased after activation of the 26S proteasome by FAT10 and NUB1L (Fig 1G). To distinguish, whether the measured effect was due to binding of FAT10 or also required degradation of FAT10, we used the slowly degradable stabilized FAT10 mutant HA–FAT10c0c134L for activation (Fig 1H) (Aichem et al, 2018). FAT10c0c134L on its own was not able to activate the 26S proteasome and caused even a slight inhibition of the proteasome activity. However, together with NUB1L it led to an increased Z–GGL–AMC cleavage, suggesting that the activation of the 26S proteasome is a gate-opening process triggered by the binding of FAT10 and NUB1L to the 19S regulator, rather than an effect on the catalytically active site subunits β1, β2, and β5 of the 26S proteasome.

### The activation of the 26S proteasome by FAT10 and NUB1L is highly specific and relies on the UBL domains of FAT10 and the UBL and UBA domains of NUB1L

Having shown an activation of the 26S proteasome by NUB1L and monomeric FAT10, we aimed to investigate whether FAT10ylated proteins can activate the 26S proteasome as well. Therefore, we used the UBA6-specific E2 enzyme USE1, the FAT10ylation of which has been documented in detail (Aichem et al, 2010, 2014). After in vitro generation of a branched USE1–FAT10 conjugate, as described in the methods section, FAT10ylated USE1 was used in an activity assay. Neither USE1 alone nor the USE1–FAT10 conjugate alone were capable to activate the 26S proteasome. The FAT10ylated USE1 activated the 26S proteasome only together with NUB1L (Fig 3A). This is a clear difference to ubiquitylated proteins, which, like poly-ubiquitylated E6AP, do not require a cofactor for activation. To investigate the specificity of this joint activation of the 26S proteasome by FAT10 and NUB1L, we tested if other reported

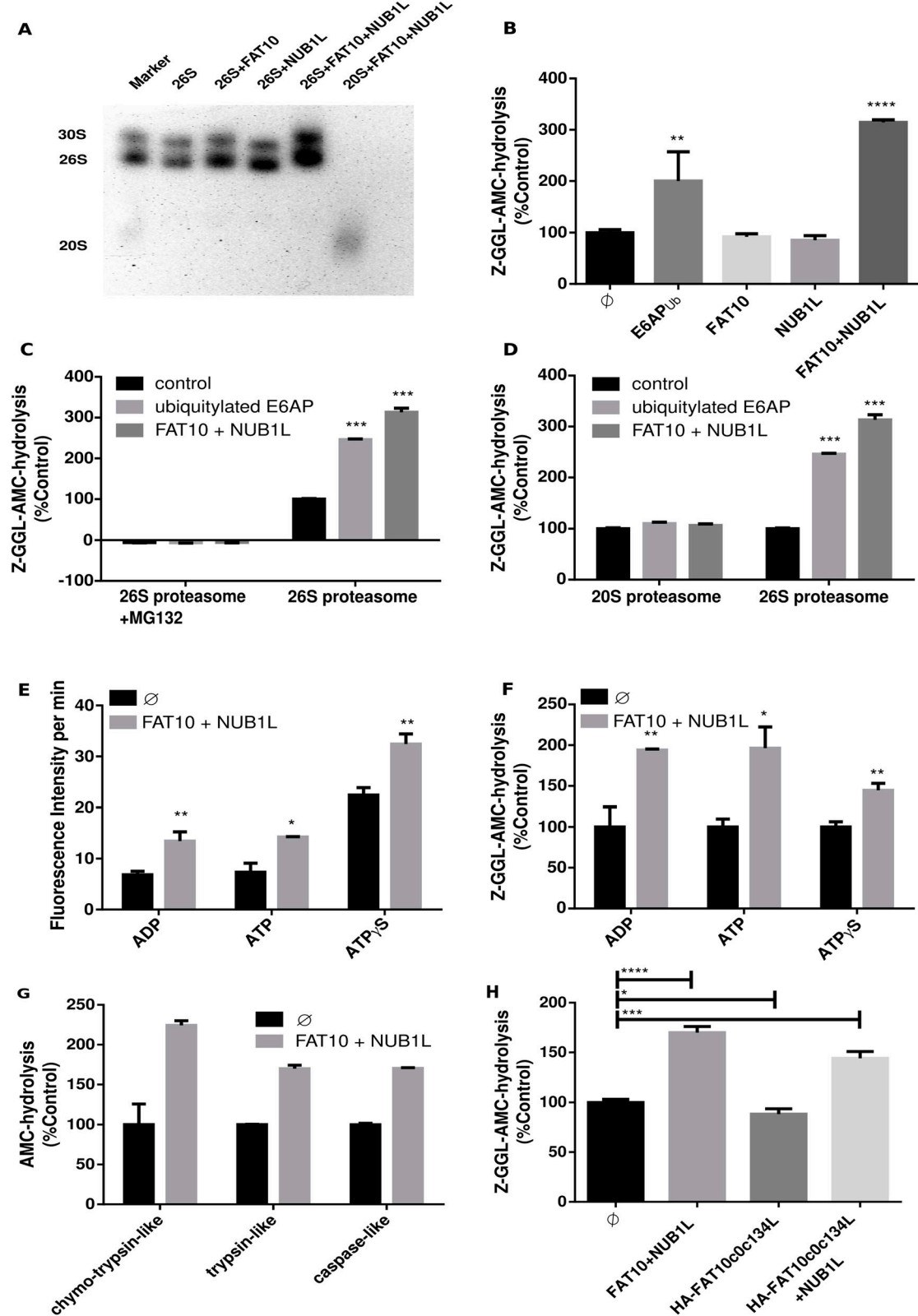

**Figure 1. Activation of the 26S proteasome by FAT10 and NUB1L.**
**(A)** A native gel overlay assay was performed with human 26S proteasome in presence or absence of FAT10 and NUB1L. Samples were initially incubated for 15 min at 37°C with a 10-fold molar excess of NUB1L and FAT10 as compared to the 26S proteasome, before gel electrophoresis was performed. The overlay assay was performed with Suc–LLVY–AMC for 30 min at 37°C. The gel was then developed under UV light. The marker was a mix out of 20S and 26S proteasomes. **(B)** Z–GGL–AMC hydrolysis was

binding partners of NUB1L could lead to a similar effect. First, the interaction of NEDD8 as an ubiquitin-like modifier with NUB1L was reported several years ago (Kito et al, 2001). Similar to what we had found for FAT10, the binding of NUB1 was reported to lead to accelerated degradation of NEDD8 (Kamitani et al, 2001; Liu et al, 2013; Tanji et al, 2019). Second, α-synuclein was found to bind to NUB1 in Lewy bodies together with its interaction partner synphilin-1. Neither NEDD8 nor α-synuclein could activate the 26S proteasome on their own or in presence of NUB1L (Fig 3B). This specificity could relate to the multiple interaction sites between NUB1L, FAT10, and the 19S regulator. Therefore, we used NUB1L mutants lacking either the UBL-domain (NUB1LΔUBL) which is essential for RPN1 and RPN10 binding, or a NUB1L variant lacking all three UBA domains (NUB1LΔUBA), as the N-terminal UBL domain of FAT10 binds to the UBA domains of WT NUB1L (Schmidtke et al, 2006). In the performed activity assay, we observed that an accelerated AMC-release from Z–GGL–AMC was only achieved by FAT10 together with full-length WT NUB1L but not with NUB1LΔUBL or NUB1LΔUBA (Fig 3C). Similarly, the deletion of one of the two UBL domains of FAT10 resulted in loss of 26S proteasome activation (Fig 3D). Although we already had used FAT10ylated proteins to activate the 26S proteasome and detected activation with the stabilized FAT10c0c134L variant, we asked ourselves if the stability of a FAT10 fusion protein would have any influence on its interaction with the ATPase subunits of the base complex of the 19S regulator. Yu et al (2016) had reported that mere binding of substrate proteins via UBL domains to the 26S proteasome is not sufficient for degradation but that an unstructured initiation sequence is required, which is grasped by the ATPase subunits of the 19S regulator and used to pull the substrate into the 20S core particle. FAT10–GFP fusion proteins were not analyzed by Yu et al (2016). In contrast to other UBL–GFP fusion proteins, FAT10–GFP had been shown to be degraded with a half-life of about 3 h (Hipp et al, 2005), whereas RAD23–UBL–GFP was stable (Yu et al, 2016). The degradation of FAT10 and its conjugates may be much more dependent on the N-terminal unstructured region of FAT10, as earlier reported by Aichem et al (2018). To investigate if an unstructured region at the FAT10 C-terminus would have an influence on the activation of the 26S proteasome, we used two recombinant FAT10 fusion proteins, FAT10–GFP and FAT10–GFPcytb (Fig 3E). The Cytb polypeptide has been described to be unstructured and to cause the difference between fast and slow degradation of Cytb-containing fusion proteins (Yu et al, 2016). We were able to see approximately the same activation of the 26S proteasome by both, FAT10–GFP and FAT10–GFPcytb, in combination with full-length NUB1L (Fig 3E). FAT10 alone is degraded about two times faster than FAT10–GFP (Hipp et al, 2005); however, both activated the 26S

proteasome to the same extend, and we concluded that the speed of degradation is not important for the activation or can cause a difference in activation. Furthermore, an unstructured region at the C-terminus of FAT10 fusion proteins seems not to have an impact on the 26S proteasome activation by FAT10. The NUB1L–UBL domain by itself coupled to GFP showed no activating effect, neither alone nor in presence of FAT10 or its recombinant fusion protein variants. Also, the separated UBL domains of FAT10 did neither activate the 26S proteasome alone nor together with NUB1L. Therefore, the UBL domains of FAT10 and NUB1L do not belong to the same category as the UBL domains of USP14, RAD23, and hPLIC 1 (Kim & Goldberg, 2018). Because both proteins, NUB1L and FAT10, can only activate the 26S proteasome as full-length proteins, apparently, the interaction of both proteins might be necessary. A hypothetical implication of these results could be a required linkage of RPN1 and RPN10 by NUB1L and bound FAT10. To test this hypothesis, we generated a NUB1LΔUBA–p21–FAT10 fusion protein. This was supposed to mimic FAT10 bound to NUB1L, and the poorly folded p21 moiety should provide a flexible linkage of appropriate length. However, when tested in an activity assay, the fusion protein was not able to activate the 26S proteasome (Fig 3F). Although a positive result would have supported this idea, we nevertheless do not want to completely rule out the possibility that a crosslinking of RPN1 and RPN10 can occur.

## Binding of FAT10 to NUB1L enhances its affinity for proteasome subunit RPN1

As we demonstrated, FAT10 and NUB1L can activate the 26S proteasome only as full-length proteins, giving first indications to a possible mechanism. USP14 serves as a receptor protein for ubiquitylated proteins at the 19S regulator, leading to an open-gate state of the 26S proteasome (Peth et al, 2009), either by binding via its UBL or USP domain (Aufderheide et al, 2015; Kim & Goldberg, 2018). To investigate the importance of USP14 in our purified 26S proteasome preparations, we performed Western blots after running native acrylamide gels. In these experiments, we used the affinity purified 26S proteasome alone, 26S proteasome incubated with NUB1L, and the 26S proteasome together with NUB1L and FAT10. Although USP14 was prominently bound to the 26S proteasome, its abundancy decreased after incubation of the 26S proteasome with NUB1L and was absent from the 26S proteasome after incubation with NUB1L and FAT10. In addition, we could see an upshift in the band of 26S-bound NUB1L when FAT10 was present (Fig 4A). Apparently, FAT10 has a recruiting function for NUB1L to the 26S complex, and the presence of USP14 is not required for this

monitored in an activity assay. Equal amounts of 26S proteasome (5 nM) were incubated with an excess of NUB1L and FAT10 (500 nM) or ubiquitylated E6AP. The chymotrypsin-like activity was measured with Z–GGL–AMC (10 µM) at 37°C. The fluorescence intensity was measured over a period of 90 min. Proteasomal activity was expressed in relation to 26S proteasome without any stimulant. **(C)** The activity assay was performed as in (B) but in the presence or absence of 1 µM proteasome inhibitor MG132. **(D)** The activity assay, as described in (B), was performed with equal amounts of 20S and 26S proteasomes (5 nM). **(E)** The activity assay as described in (B) was performed with equal concentrations of ADP, ATP, and the non-hydrolysable ATPγS (0.5 mM) in the presence or absence of FAT10 and NUB1L. **(F)** Shows the same experiment as performed in (E) but the percentage of activity was always calculated between the corresponding control (black) and the 26S proteasome in presence of FAT10 and NUB1L (grey). **(G)** The activity assay was performed as described in (B) but in addition to Z–GGL–AMC (10 µM) to determine the chymotrypsin-like activity, the trypsin-like activity with LRR–AMC (10 µM), and the caspase-like activity with Ac–nLPnLD–AMC (10 µM; nL = norleucine) was measured. **(H)** The activity assay was performed as described in (B) using the native FAT10 and the stabilized form of FAT10, FAT10c0c134L, in the presence or absence of NUB1L. Statistical analysis was performed using an unpaired t test (ns, not significant: P > 0.05, *P ≤ 0.05, **P ≤ 0.01, ***P ≤ 0.001, and ****P ≤ 0.0001).
Source data are available for this figure.

**A**

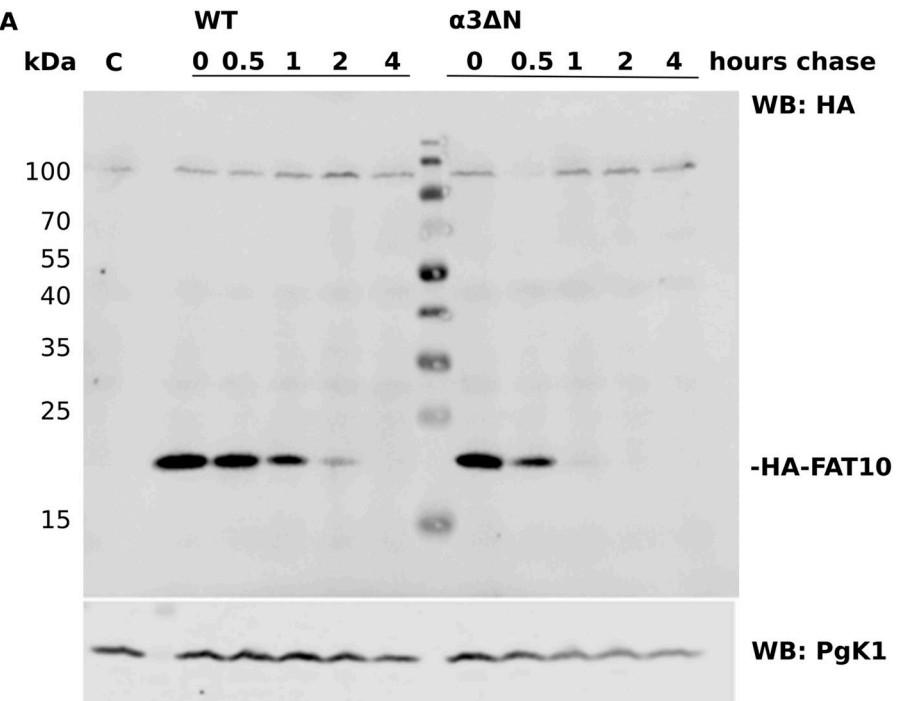

**Figure 2. The degradation of FAT10 by open gate and WT proteasome.**
**(A)** A Western blot was performed with samples of two cycloheximide chase experiments. On the left side, WT *S. cerevisiae* was transfected with HA–FAT10 Met vector, and the degradation of FAT10 was monitored over the period of 4 h. On the right side, the 20S proteasome open-gate yeast mutant α3ΔN was transfected with the same vector. The degradation of HA–FAT10 was increased because of the open-gate state of the 26S proteasome. **(B)** The experiment is the same as in (A), except for the co-transfection of both yeast types with HA–NUB1L in a Leu vector in addition to the HA–FAT10 Met vector. In the open-gate mutant, the degradation of HA–FAT10 was not accelerated by NUB1L in contrast to the WT yeast (when compared with HA–FAT10 degradation in panel (A)).

**B**

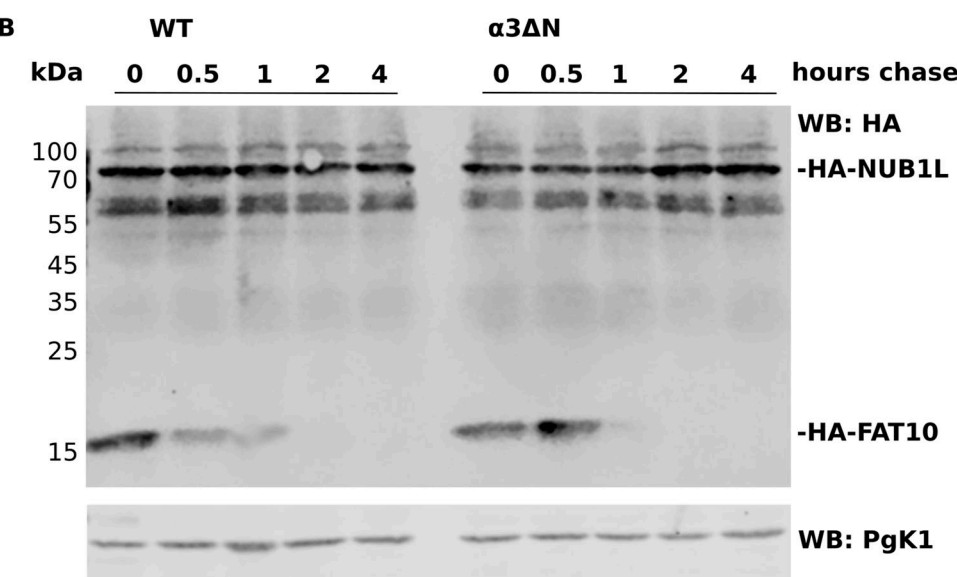

function. Hence, we performed affinity measurements with an Octet instrument for the quantitative assessment of the affinity of NUB1L and FAT10 for RPN1 and RPN10, respectively. His6–RPN1 was loaded on the sensor of an Octet instrument, and association and dissociation of NUB1L was measured. Although GST–FAT10 did not detectably bind to RPN1 in accordance with a previous report (Rani et al, 2012), it enhanced the affinity of NUB1L to RPN1 about fourfold (Fig 4B). The negative control protein GST, when co-incubated with NUB1L, did not increase the affinity of NUB1L for RPN1 at all.

Moreover, the affinity of FAT10 towards RPN10 did not change after co-incubation with NUB1L (Fig S1). Next, we investigated whether the interaction of FAT10 and NUB1L enabled the UBL domain to bind to RPN1. One of our hypotheses was the formation of NUB1L dimers occurring in the absence of FAT10. To test this possibility, we performed a GST pulldown with GST–NUB1L and a constant amount of FAT10–GFP–Cytb and an increasing amount of NUB1L–UBL–GFP–Cytb or vice versa (Fig 4C). Although bound FAT10 was still visible at a concentration ratio from 1:4, the NUB1L–UBL–GFP–cytB

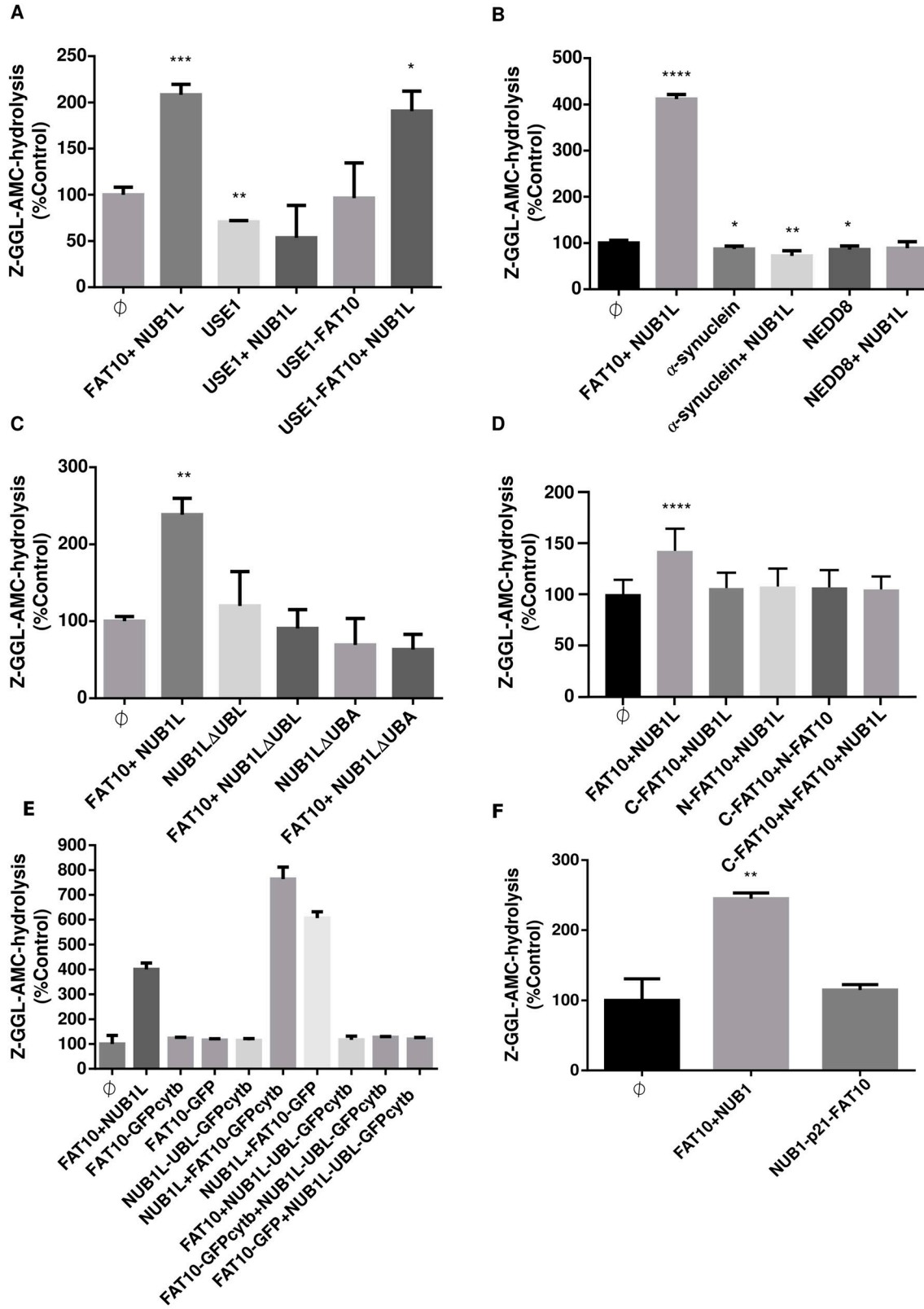

**Figure 3. Dissection of domains in FAT10 and NUB1L for activation of the 26S proteasome.**
**(A)** The activity assay was performed as described in Fig 1B. In addition, the 26S proteasome was incubated with equal amounts of USE1 and the USE1–FAT10 conjugate (500 nM). Only FAT10 and FAT10ylated USE1 together with NUB1L led to an increase in proteasomal activity. **(B)** Equal amounts of indicated known NUB1L interaction partners (500 nM) were incubated with NUB1L and the 26S proteasome. Only FAT10 showed the activating effect together with NUB1L. **(C)** Recombinant NUB1L variants

protein was only visible in the pulldown at a ratio of 1:2. It emerged that the UBL domain of NUB1L binds to its own UBA domains but that FAT10 has a slightly higher affinity for them.

### FAT10 and NUB1L activate the 26S proteasome independently of the presence of USP14

To investigate a possible competition between USP14 and NUB1L binding to RPN1, we performed another pull-down experiment. We used GST–RPN1 bound to GSH beads and performed a pulldown with NUB1L alone (Fig 5A, left-most lane). Then, we used increasing concentrations of USP14 to test a potential competitive binding to RPN1. Although we observed a slight increase for RPN1-bound USP14 in the pull-down with increasing amounts of USP14 (Fig 5A, left panel), the amount of bound NUB1L did not change with different concentrations of USP14. Under the influence of FAT10 (Fig 5A right panel), binding of USP14 to RPN1 did not change with increasing amounts of USP14, whereas the binding of NUB1L slightly increased in agreement with data shown in Fig 4B. These experiments strengthened our hypothesis that USP14 is not necessary for the activation of the 26S proteasome by FAT10 and NUB1L.

During previous in vitro degradation assays, we observed that ubiquitylated proteins were degraded slower after the addition of higher than stoichiometric amounts of FAT10 and NUB1L, as compared to the 26S proteasome alone. We used the short-lived poly-ubiquitylated protein SIC1 as 26S degradation substrate in vitro and took aliquots every 5 min. In a Western blot, we could show its degradation by the purified 26S proteasome indicating its functional integrity, but in the presence of FAT10 and NUB1L, the degradation of poly-ubiquitylated SIC1 was retarded (Fig S2A). Similar results were obtained with radioactively labeled poly-ubiquitylated p53 (Fig S2B).

Finally, we posed the question of whether USP14 is required for the 20S gate opening and activation of the 26S proteasome by FAT10 and NUB1L, as it has been shown for poly-ubiquitylated proteins (Peth et al, 2009). For this purpose, we used a USP14$^{-/-}$ MEF cell line created from USP14-deficient mice (Lee et al, 2010). We purified the 26S proteasome from the USP14$^{-/-}$ cell line and from a MEF WT control cell line (C4). The purified 26S proteasomes were tested for the presence of USP14, which is commonly co-purified with the 26S proteasome because of its non-covalent binding to the 26S complex. 26S proteasome affinity purified from human erythrocytes and from C4 WT control cells contained USP14, whereas USP14 was not detectable in the purified 26S proteasome from USP14$^{-/-}$ cells (Fig 5B). In an activity assay with 26S proteasome from C4 cells and USP14$^{-/-}$ MEFs, we were able to show similar 26S activation by FAT10 and NUB1L (Fig 5C). In conclusion, the 20S gate opening and activation of the 26S proteasome by FAT10 plus NUB1L did not rely on

the induction of a conformational change or displacement of USP14 from the 26S complex as previously demonstrated for poly-ubiquitylated proteins and hence constitutes a new mechanism of 26S activation (Fig 5D).

## Discussion

The degradation of proteins by the 26S proteasome is a strictly regulated process. One of the best investigated pathways is the 26S-mediated degradation of poly-ubiquitylated proteins (Bard et al, 2019). These covalently conjugated proteins are recognized by receptor subunits of the 19S regulatory particle which leads to their insertion into the core particle. These different states of the 26S proteasome were characterized as the substrate-accepting state (s1), the intermediate state (s2), and the translocating state (s3) (Matyskiela et al, 2013; Unverdorben et al, 2014). In the s1 state, poly-ubiquitylated proteins can bind to the receptors RPN10, RPN13, and RPN1 of the 19S regulator particle. The s2 state enables the functionality of the DUB RPN11, which removes ubiquitin chains. In the s3 state, the ATPases of the 19S regulator insert with their HbYX motif between the 20S α subunits and lead to an open-gate of the 20S core particle (Smith et al, 2007), which allows the translocation of proteins into the core particle. This change enables the enhanced capability of hydrolysis of small peptides by the core particle. This capability is also reached in the 20S proteasome, which lacks the N-terminal domain in its α subunits (Groll et al, 2000). A DUB-designated USP14 in mammals was reported to play a pivotal role as an activator of the 26S proteasome. In experiments performed by the Goldberg group (Peth et al, 2009), the facilitated hydrolysis of peptides by USP14 was enabled by induction of a conformational switch induced by non-covalent binding of poly-ubiquitylated proteins to USP14 leading to maximal gate opening in the 20S core particle (Peth et al, 2009). It was also shown that open-gate mutants lack further activation by USP14. Furthermore, the activation of the 26S proteasome was linked to conjugates bearing ubiquitin chains. Monomeric ubiquitin did not enhance peptide hydrolysis.

In this study, we investigated whether FAT10, serving as a second direct degradation tag within the ubiquitin modifier family, similarly activates the 26S proteasome. We found that FAT10 can serve this activation function as well, but needs its interaction partner NUB1L to achieve this goal. Ubiquitin and FAT10 are both members of the same protein family, but their binding sites and structure differ. For example, the ubiquitin receptor RPN10 is binding poly-ubiquitin at its UIM domains and FAT10 at its VWA domain (Elsasser et al, 2004; Rani et al, 2012). The FAT10 interaction partner NUB1L can bind to both, RPN10 and RPN1 (Rani et al, 2012), and facilitate the

---

lacking either the three UBA or the UBL domain were incubated with the 26S proteasome in the presence or absence of FAT10. Only full-length NUB1L had the ability to lead to an activation of the 26S proteasome. **(D)** The two FAT10–UBLs were incubated alone or together with NUB1L. Only full-length FAT10 was able to activate the 26S proteasome together with NUB1L. **(E)** Recombinant NUB1L–UBL–GFP–cytb containing the UBL domain of NUB1L only, FAT10–GFP and FAT10–GFP–cytb fusion proteins were tested in equal amounts for their ability to activate the 26S proteasome, as indicated. Although conjugated FAT10–GFP–Cytb and FAT10–GFP are both suitable to activate the 26S proteasome, the UBL domain in NUB1L–UBL–GFP–Cytb is not enough for the activation. **(F)** A NUB1–p21–FAT10 fusion protein was incubated with the 26S proteasome to test whether NUB1L and FAT10 need to be covalently linked, for example, by a flexible linker (p21) for activation of the 26S proteasome. Activation by FAT10 and NUB1L served as positive control. Statistical analysis was performed using an unpaired t test (*P ≤ 0.05, **P ≤ 0.01, ***P ≤ 0.001, and ****P ≤ 0.0001). Source data are available for this figure.

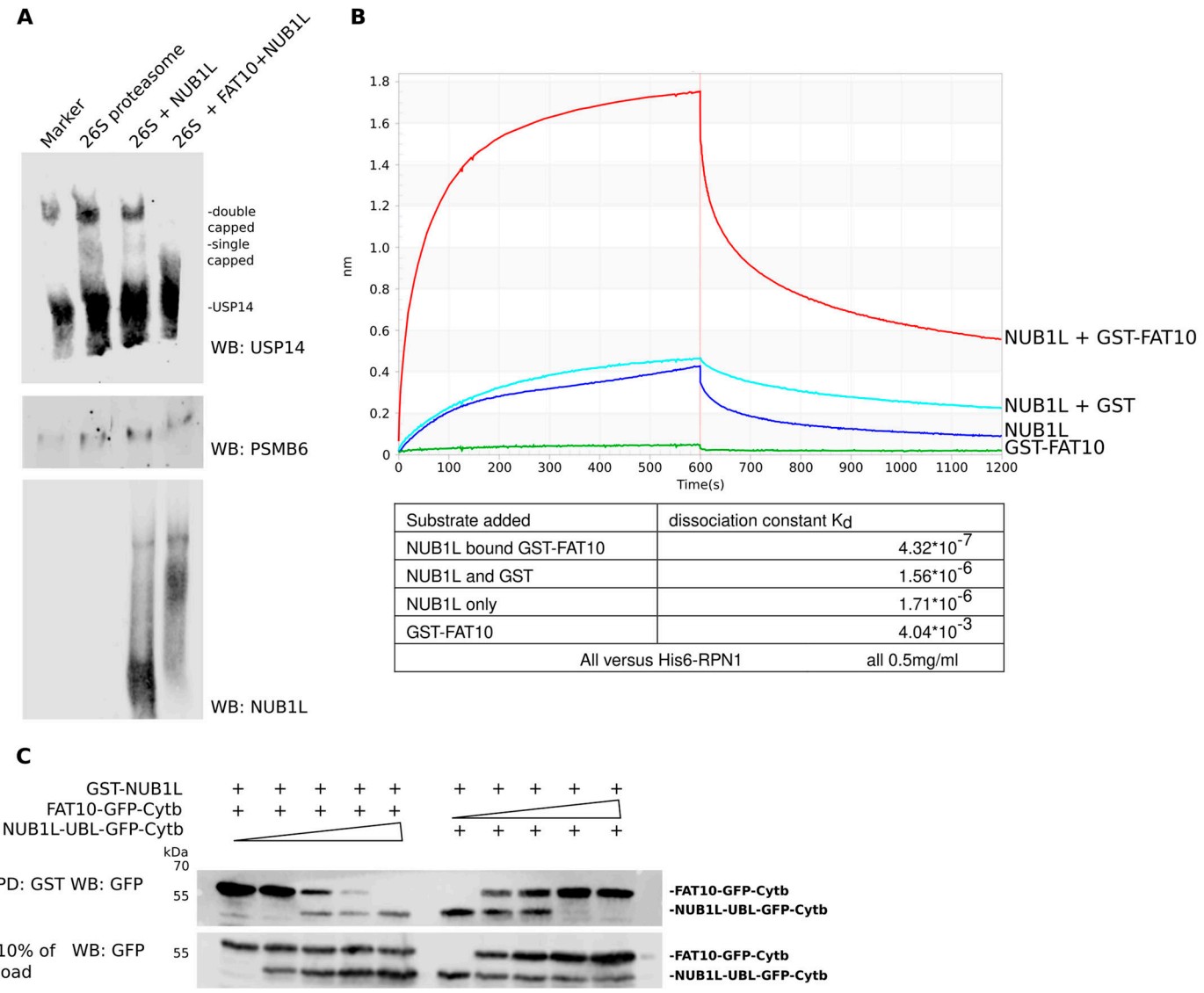

**Figure 4. FAT10 increases the affinity of NUB1L for binding to the 26S proteasome.**
**(A)** Western blot of a native gel, displaying the interaction of FAT10 and NUB1L with the 26S proteasome. NUB1L alone, and FAT10 and NUB1L together are capable of interfering with the non-covalent binding of USP14 to the 26S proteasome. **(B)** Octet-binding curve of NUB1L-bound GST–FAT10 (red), NUB1L and GST (light blue), NUB1L only (dark blue), and GST–FAT10 (green) for membrane-bound His6–RPN1. The binding of NUB1L to RPN1 increases markedly when GST–FAT10 is bound His6–RPN1 as is also documented by the calculated dissociation constants $K_d$ shown in the table below. **(C)** A pulldown (PD) was performed with GST–NUB1L. On the left side, FAT10 was incubated with GST–NUB1L and competed with increasing amounts of the UBL domain of NUB1L, fused to GFP–Cytb (named as NUB1L–UBL–GFP–Cytb): lane 1, 1:0; lane 2, 1:1; lane 3, 1:2; lane 4, 1:4; and lane 5, 1:8. At a ratio of 1:2, the NUB1L–UBL–GFP–Cytb was able to bind to NUB1L and to displace FAT10–GFP–Cytb. On the right side, the experiment was changed and NUB1L–UBL–GFP–Cytb had to compete with increasing concentrations of FAT10. In this case, a 1:1 ratio was sufficient to displace NUB1L–UBL–GFP–Cytb from GST–NUB1L and results in both proteins being bound to GST–NUB1L.

degradation of FAT10 (Hipp et al, 2004). During our experiments, we investigated the ATP dependency of the increased proteolysis induced by FAT10 and NUB1L. We assumed that an activation of proteasomal activity by a gate-opening effect of the core particle mediated by the 19S regulatory particle would require ATP, as reported earlier (Benaroudj et al, 2003; Smith et al, 2005; Rabl et al, 2008; de la Pena et al, 2018). We were able to show that the effect of increased peptide hydrolysis in activity assays by FAT10 and NUB1L was decreased when performed with ATPγS, which, as a slowly hydrolyzing variant of ATP, leads to a preferred open gate state of the 26S proteasome (Smith et al, 2005). Because the entry of small

peptides in the 20S proteasome is the rate-limiting step for their degradation (Kisselev et al, 2002), the gate opening of the 20S could also be a suitable explanation for the increased activity of all three peptidase sites for small peptides. Evidence for this hypothesis was obtained by experiments using an open-gate mutant yeast proteasome. FAT10 was degraded faster in these cells than in WT cells. However, the increased degradation of FAT10 in presence of NUB1L was not observed in these mutants (Fig 2). Thus, the earlier described accelerated degradation of FAT10 by NUB1L (Hipp et al, 2004) can be explained by the gate opening mediated by the FAT10–NUB1L dimer. This gate-opening effect is similar to the

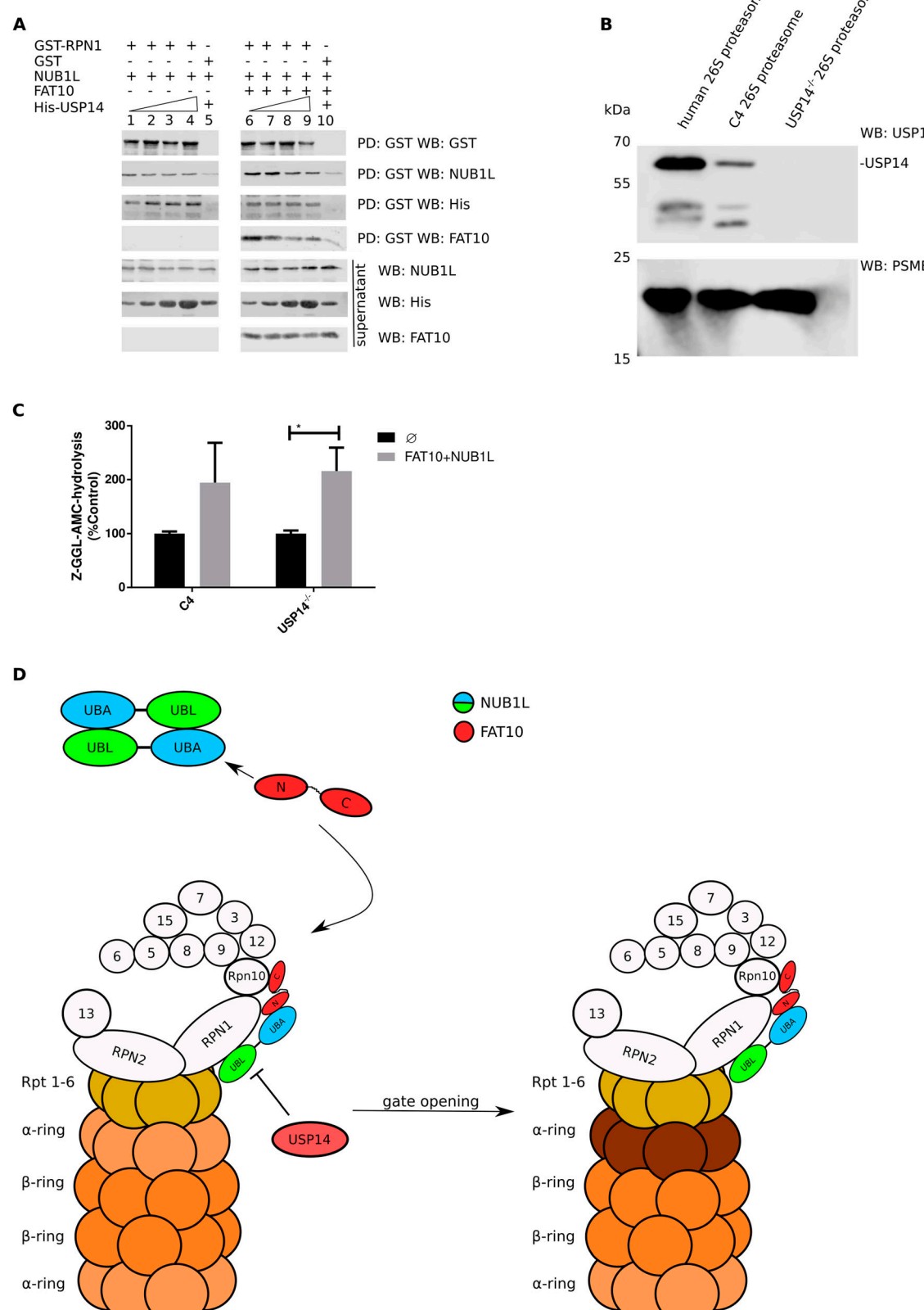

**Figure 5. USP14 independence of 26S proteasome activation by FAT10 and NUB1L.**
**(A)** A GST pulldown (PD) was performed using 20 μl GST–RPN1 incubated with ~10 ng NUB1L. NUB1L was then eluted by increasing amounts of USP14 (10 ng (1), 20 ng (2), 50 ng (3), and 100 ng (4)); in lane 5, GST served as negative control. Lanes 6–10 show the same experiment with additional 5 ng FAT10. An increase of bound NUB1L to RPN1 is visible in presence of FAT10 (compare lanes 1–4 to 6–9, second gel from top: IP: GST, WB: NUB1L). **(B)** The USP14 WB shows the abundance of USP14 in purified 26S proteasome from erythrocytes, the mouse

results described earlier by Bajorek and colleagues, for the activation by ubiquitylated proteins (Bajorek et al, 2003).

Because the degradation of FAT10 was already described to be increased up to eightfold in presence of NUB1L ([Hipp et al, 2004] and also illustrated in Fig 2), we were concerned that the gate opening of the 26S proteasome is an opening due to FAT10 degradation rather than a specific mechanism for gate opening transmitted by 19S subunits. To address this possibility, we used the stabilized version of FAT10, FAT10c0c134L, which is degraded slower than WT FAT10 (Aichem et al, 2018), and thus favors binding to 19S subunits over the degradation of FAT10. The activation of the proteasome by FAT10c0c134L was an indicator that the activation of the 26S proteasome relies more on the binding of FAT10 than its continuous degradation. As both, FAT10 and a FAT10–GFP fusion protein, together with NUB1L activated the 26S proteasome to the same extent, but differed in their half-life by more than a factor of two, this further supported our hypothesis that the activation depends on the binding rather than on the degradation. In contrast to unconjugated monomeric and poly-ubiquitin (Peth et al, 2009), monomeric unconjugated FAT10 is an efficient activator of the 26S proteasome, however, only together with NUB1L. Although the activation of the proteasome is not limited to monomeric FAT10 and can also be accomplished by recombinant, branched FAT10 conjugates, the activation required full-length FAT10 and NUB1L protein (Fig 3). Mono-ubiquitylated proteins do not lead to activation (Peth et al, 2009). Furthermore, they need the DUB USP14 to activate the 26S proteasome. The question to be asked was how the interaction between FAT10, NUB1L, and the 26S proteasome leads to increased peptidase activity. Previous studies of our group indicated two possible ways of FAT10 and NUB1L binding to the 26S proteasome. Although FAT10 binds via its C-terminal UBL-domain to the von Willebrand A (VWA) domain of RPN10, the UBL domain of NUB1L can bind to both, RPN10 and RPN1. Because the N-terminal UBL domain of FAT10 is bound by the UBA domains of NUB1L, FAT10 can bind RPN10 either directly or is transported indirectly via NUB1L to RPN10 (Rani et al, 2012). In our experiments, we were able to show that in vitro NUB1L formed dimers and is binding to itself via its UBA and UBL domain and that FAT10 competes with the UBL domain of NUB1L for the binding of the three UBA domains of NUB1L, resolving the dimer/folding and therefore enabling the UBL domain of NUB1L to bind to RPN1 (Fig 4C). We were able to show that there is an increased binding affinity of NUB1L to RPN1 in presence of FAT10 (Fig 4B). The Western blot of our native gel (Fig 4A) and the binding experiment of USP14 and NUB1L (Fig 5A) showed an increased affinity of RPN1 for NUB1L in presence of FAT10, as well. Because both USP14 and NUB1L bind to RPN1 and thereby are situated in the vicinity of gate-opening ATPases, we could assume gate-opening effects that are targeted by RPN1, especially because the gate-opening effect of USP14 was already shown in cryo-electron microscopy with the yeast homolog UBP6 (Aufderheide et al, 2015). Of note, only a small amount of proteasomes in the cells contain

USP14. Despite its low dissociation constant of about 4 nM in the presence of Ub-AMC (Lee et al, 2010), USP14 exists not only bound to proteasomes but also unbound. In a publication from Kuo and Goldberg (2017), it was reported that poly-ubiquitin enhances the affinity of USP14 to the proteasome, and that USP14 dissociates after degradation of the poly-ubiquitin substrate. Therefore, a competition between FAT10 and NUB1L on one site and USP14 on the other site may not occur in vivo, as there is large amount of USP14-free proteasome.

Although NUB1L needs its binding partner FAT10 to efficiently bind to RPN1, we were not able to observe a different binding capacity for FAT10 to RPN10 with or without NUB1L (Fig S1). These findings, together with the findings that both proteins, unlike other UBL proteins, are needed as full-length proteins to activate the 26S proteasome, led to the question if FAT10 and NUB1L serve as a cross-linker between RPN1 and RPN10, as it is possible with poly-ubiquitin. However, our first experiments with a recombinant linker protein NUB1L–P21–FAT10 did not show an activation of the 26S proteasome. P21 was used as a short flexible linker, but more experiments with several linker lengths must be conducted to proof a link between RPN1 and RPN10 for FAT10/NUB1L-mediated activation. It was shown that not only the UBL domain of USP14 leads to increased peptide hydrolysis but also the UBL domains of RAD23 A and B were able to activate the 26S proteasome (Kim & Goldberg, 2018). In the same article, it was shown that such an activation leads to increased degradation of poly-ubiquitylated proteins by the UBL domain of RAD23 A and B. However, this effect was only observed if low amounts of DNA (0.5 µg) were transfected. Transfection of higher amounts (1 or 2 µg) led to an accumulation of poly-ubiquitylated proteins, the opposite of one would expect an activation. We therefore suggest that too much of the activator somehow blocks the access of the substrates. A comparable result was published by Zhang et al (2003). The authors published that ODC and antizyme are degraded slower by the 26S proteasome in the presence of poly-ubiquitin. If poly-ubiquitin would always accelerate the degradation, one would expect a faster degradation of ODC–antizyme. Here, the authors suggested a competition between ODC, antizyme, and poly-ubiquitin. Later, Beenukumar Renukadevi (2015) showed that mutation or deletion of ubiquitin receptors did not prevent the degradation of ODC and antizyme, so that the possibility of a competition can rather be excluded, and the results by Zhang et al (2003) can be explained by an overload of the system. A UBL dependent degradation similar to that of RAD23 was shown for many other UBL domains (Yu et al, 2016; Collins & Goldberg, 2020); however, the UBL domain of NUB1L was not investigated in these publications. We observed a diminished degradation of ubiquitylated proteins upon the addition of NUB1L and FAT10 in vitro (Fig S2). Furthermore, the ubiquitin smear did not change upon tetracyclin-induced expression of mouse FAT10 in mouse fibroblasts and transiently overexpressed FAT10 in human HEK293 cells (Raasi et al, 2001). Reports about an accelerated degradation of poly-ubiquitylated proteins in vitro,

fibroblast line C4, and the USP14 knockout mouse fibroblast line. PSMB5 was used as loading control. **(C)** A fluorogenic peptide (Z–GGL–AMC) based activity assay was performed as described in the legend to Fig 1B. The 26S proteasomes purified from the USP14 proficient fibroblast line C4 and a USP14 deficient fibroblast line were equally activated by FAT10 + NUB1L. Statistical analysis was performed using an unpaired t test (*P ≤ 0.05). **(D)** Hypothetical scheme in which FAT10 resolves NUB1L dimers by binding to the UBA domains of NUB1L. Thereby, the UBL domain of NUB1L gets free to bind to RPN1 and replace USP14. Consequently, the C-domain of FAT10 is free to bind to RPN10 and activate the 20S gate opening. Source data are available for this figure.

induced by any UBL is lacking so far, and in vivo the activation of the 26S proteasome was observed for one specific concentration only (Collins & Goldberg, 2020). We therefore suggest that we might not have used those concentrations of our activators, required to observe a faster degradation of poly-ubiquitylated substrates. Another explanation might derive from the different binding mode of NUB1L to RPN1. Whereas poly-ubiquitylated proteins and RAD23 bind to the T1 site of RPN1, USP14 binds to the T2 site (Shi et al, 2016). Recent Octet and pull-down experiments in our group have shown that human RAD23 binds exclusively to the T1 toroid of RPN1 as shown before in *S. cerevisiae* (Shi et al, 2016) and that NUB1L binding gets less after mutating both, the T1 and the T2 site of human RPN1 (Fig S3), NUB1L still binds to RPN1 if both sites are mutated. This may explain why the UBL domains of NUB1L alone and FAT10 alone cannot activate, in contrast to RAD23 or USP14. This interaction of NUB1L could explain the decreased degradation of poly-ubiquitylated proteins. Furthermore, the binding of FAT10 at RPN10 could interrupt the downstream interaction of the RPN10 VWA domain after RAD23 interaction with RPN1 (Verma et al, 2004; Shi et al, 2016). The competition on two poly-ubiquitin binding sites at the proteasome leads to the assumption that FAT10ylated proteins are favored for degradation. Similar behavior of competition for the open-gate proteasome was seen by the ubiquitin-independent route of ornithine decarboxylase (ODC) and its accelerator antizyme 1 (AZ1) and poly-ubiquitin (Zhang et al, 2003). In these experiments the initial binding step of poly-ubiquitin was able to inhibit the degradation of ODC in vitro. This result was interpreted as competition of these two proteins for one binding site. Although this thesis works well for the competition of FAT10/NUB1L and poly-ubiquitin with its shuttle factors, Beenukumar et al showed that ODC has specific receptors at the proteasome (Beenukumar et al, 2015). Considering all new findings in this article, we report on a new USP14-independent activation mechanism of the 26S proteasome facilitated by NUB1L and FAT10. These two IFN-γ and TNF-α inducible proteins are activating the proteasome by the binding of the N-domain of FAT10 to the UBA domains of NUB1L and thereby dissolving NUB1L dimers. This results in a free UBL domain of NUB1L which has a high affinity to the RPN1 subunit of the 19S regulator. The free C-domain of FAT10 can bind to RPN10 and thereby leads to a gate opening of the 20S core particle (see scheme in Fig 5D). This gate opening is ubiquitin and USP14 independent and our data might indicate a favored proteasomal degradation for FAT10ylated proteins because a decreased degradation of poly-ubiquitylated proteins was observed in the presence of FAT10 and NUB1L (Fig S2). A preferential degradation of FAT10ylated proteins might be required to enable the rapid disposal of FAT10ylation substrates during infection or inflammation.

# Materials and Methods

### Cell lines

HEK293 human embryonic kidney cells, C4 mouse fibroblasts, and USP14$^{-/-}$ mouse fibroblasts were cultivated in DMEM (Thermo Fisher Scientific) supplemented with 10% fetal calf serum (Gibco/Thermo Fisher Scientific), 1% stable glutamine (100×, 200 mM), and 1% penicillin/streptomycin (100×) (both from Biowest/VWR). The C4 murine fibroblast line was derived from embryonic BALB/c mice by SV40 infection in vitro and is described in reference (Schwarz et al, 2000). The mouse USP14$^{-/-}$ fibroblasts were generated from USP14$^{-/-}$ mice (Crimmins et al, 2009), as previously described (Lee et al, 2010); the cells were a kind contribution from Daniel Finley (Harvard University).

### Expression and purification of GST–UBL

pDEST15–UBL–hHR23B was transformed into B834 *E. coli* (Novagen) according to manufacturer's instructions. The bacteria were grown at 37°C to an $OD_{600nm}$ of 0.6 and induced for 3 h with 1 mM IPTG (Roth). Pellets of 1 liter bacteria culture were then lysed in 25 ml GSH-binding buffer (GBB) (1x PBS, 10 mM $MgCl_2$, 1 mM DTT) by sonication. Afterwards, the bacteria were centrifuged at 100,000$g$ for 1 h. 500 $\mu$l glutathione-Sepharose (GE Healthcare) was added to the supernatant for batch purification. After washing with 75 ml GBB, the beads were stored for further use.

### Expression and purification of His$_{10}$–UIM

pET26b–His10–UIM2 was transformed into B834 *E. coli* (Novagen) according to the manufacturer's instructions. The bacteria were grown at 37°C to an $OD_{600}$ of 0.6 and induced for 3 h with 1 mM IPTG (Roth). Pellets corresponding to 2 liters bacteria culture were lysed in 25 ml lysis buffer (50 mM $NaH_2PO_4$, 300 mM NaCl, pH 8.0, using NaOH) by sonication. Thereafter, the lysed bacteria were centrifuged at 100,000$g$ for 1 h. Afterwards, 0.5 ml of Ni–IDA resin (Protino) were added to the supernatant for batch purification. After washing with 75 ml of lysis buffer, the protein was eluted by adding 2 ml elution buffer (50 mM $NaH_2PO_4$, 300 mM NaCl, pH 8.0, 300 mM imidazole) for 2 h hours at 4°C while rotating. The imidazole concentration was then reduced to 0.1 mM by ultrafiltration (Ultracel-3K; Merck). The protein was then concentrated to ~2 mg/ml and stored at –20°C for further use.

### Affinity purification of the 26S proteasome

10 ml of human erythrocytes were mixed with 25 ml affinity purification buffer (25 mM Hepes–KOH, pH 7.4, 10% glycerol, 5 mM $MgCl_2$, 1 mM ATP, 1 mM DTT) left on ice for 1 h and then sonicated. After a centrifugation at 100,000$g$ for 1 h, the supernatant was added to GST–UBL bound to GSH beads. After 2 h at 4°C with rotation, the beads were washed with 100 ml affinity purification buffer. The 26S proteasome was then eluted by adding 250 $\mu$l HIS–UIM (~2 mg/ml) and rotated 4°C for 20 min. Remaining unbound HIS–UIM was removed by incubation of the supernatant of the previous step with Ni–IDA resin for additional 20 min. The 26S proteasome was then used or stored at –20°C for further use.

### Stimulation of 26S peptidase activity

Peptide hydrolysis by human 26S proteasome was measured with 10 $\mu$M GGL–AMC, LLR–AMC, or nLPnLD–AMC ($\lambda$ex = 360 nm $\lambda$em = 460 nm) (Cayman Chemical). 5 nM 26S proteasome was measured in

presence of 500 nM FAT10 or/and 500 nM NUB1L for 60 min at 37°C. The reaction mixture contained 25 mM HEPES/KOH (pH 8.0), 2.5 mM MgCl$_2$, 125 mM K-acetate, 0.025% Triton X-100, 0.5 mM ATP, 0.5 mM DTT, and 0.1 mg/ml BSA, as previously described (Peth et al, 2009).

### Native gels

Native gels with the 26S proteasome were prepared as described by Elsasser and Finley (2005) with adjustment of the incubation temperature to 37°C to match the human 26S proteasome. For experiments with FAT10 and NUB1L, the ratio between proteasome was set as in peptidase activity assays. Western blots were afterwards performed with a semidry procedure for 1 h at 8 V.

### Substrate overlay assay

The overlay assay was exactly performed as described before (Elsasser & Finley, 2005).

### SDS–PAGE

For SDS–PAGE, a 12% separation and stacking gel was used. After adding fourfold SDS sample buffer to the samples, the samples were boiled at 95°C for 5 min. Afterwards, the samples and the protein marker were applied, the gel was run for 30 min with 45 V and afterwards for 1.5 h with 110 V. The gel was stained with Coomassie dye afterwards.

### Western blot

After performing a SDS–PAGE, a Western blot was performed using 0.2 µm nitrocellulose membrane. The blot was run for 1 h 10 min with 110 V. To stain the bands two antibodies were used. The two first antibodies were reactive to β5c to stain the PSMB5 subunit and reactive to αHis6 to stain the His tag. The first antibodies were incubated overnight and washed 3x with TBS-Tween. The two secondary antibodies were anti-rabbit IgG 800 and anti-mouse IgG 680. They were incubated for 2 h, again washed 3x and the blot was imaged and quantitatively evaluated with the LI-COR Odyssey imager and the Image Studio Lite Version 5.2.

### Radioactive 26S proteasome activity test

The radioactive protein was mixed with substrate buffer and 6.32 µg 26S proteasome and in the positive control with 9.75 µg FAT10 and 34.56 µg NUB1L. The reaction was then performed at 37°C, and samples were taken at 0, 5, 10, and 20 min for poly-ubiquitylated p53 and at 0, 15, 30, and 60 min for poly-ubiquitylated E6AP. The samples were immediately boiled at 95°C with SDS–PAGE sample buffer to stop the reaction. After Western blotting, the radioactivity was measured with a radio imager (Molecular Imager FX; Bio-Rad).

### Generation of the covalent FAT10–His–USE1 conjugate

The vector pTYP2–HA–FAT10 containing human HA–tagged FAT10 as a fusion protein with intein and a chitin-binding domain was a gift from Benedict Kessler (Oxford University). All cysteines except for cysteine 134 of FAT10 were mutated to serines, Cys134 was mutated to leucine using the primers: PR3-06 SDM FAT10 INT-CBD C7S + C9S (t19a_g26c): 5′-CTCCCAATGCTTCCAGCCTCTCTGTGCATGTCCGTT-3′; PR3-08 SDM FAT10 INT-CBD C163S (t485a): 5′-CTCTTCCTGGCATCT-TATTCTATTGGATGCTTTGCC-3′; PR3-35 SDM FAT10 INT-CBD C134L (t400c_g401 t_c402g): 5′-CCCTGAAACCCAGATTGTGACTCTGAATGGAAA-GAGACTGGAAGATG-3′. FAT10 C(0)C134L protein was expressed and purified as described in the manual provided by the supplier of the chitin beads (S6651S; New England Biolabs). Cleavage from the column was prepared as previously described with 100 mM cys-tamin over night at room temperature to allow for the S to N shift. Like this, a modified FAT10 with a single SH group at the C-terminus was generated. After desalting of 500 µg of FAT10 via a PD10 column (GE healthcare) in ligation buffer (20 mM Tris, pH 7.0, 50 mM NaCl, 1 mM EDTA), a 10-fold molar excess of 5,5′- dithio-bis-(2-nitrobenzoic acid) (DTNB) was added and incubated at room temperature for 30 min. After gel filtration via a PD10 column in ligation buffer, the activated FAT10 was concentrated to 250 µl via an Amicon Ultra-0.5 filtration unit. Expression and purification of His6-tagged USE1 was performed as described in the study by Aichem et al (2010). After gel filtration via PD10 columns in ligation buffer, USE1 was concentrated to 0.5 mg/ml via an Amicon Ultra-0.5 filtration unit. 500 µl of con-centrated USE1 was mixed with 250 µl activated FAT10 (molar ratio 1:2) and incubated at room temperature. Aliquots were taken after 30-, 60-, and 120 min and analyzed by nonreducing SDS–PAGE. All USE1 was detected as a single band at 30 kD higher molecular weight as mo-nomeric USE1 already after 30 min incubation, representing the USE1-FAT10 conjugate. This conjugate was used without further purification.

The generation of tagless human FAT10 via a pSUMO vector and ULP1 digestion was performed as described before (Aichem et al, 2019). Human NUB1L and the deletion mutants delta UBA and delta UBL were cloned into the pSUMO vector via BsaI and XhoI with the primers: 5′-BsaI Nub1L 5′-CCAGTGGGTCTCAGGTGGTGCACAAAAGAAATATCTTCAAGC-3′; 3′-XhoI-Nub1L 5′-CCGCTCGAGTTAGTTTTTCTTTGTTGCTGACTTCC-3′; 3′-XhoI-Nub1L-UBA Domains 5′-CCGCTCGAGTTATCCTCCGTTGTGAGCAAGGG-3′; 5′-BsaI-Nub1L-3xUBA 5′-CCAGTGGGTCTCAGGTGGTGATCCATCAAAAGT-GGACAATTTGTTGC-3′; 5′-BsaI-Nub1L-dUBL 5′-CCAGTGGGTCT-CAGGTGGTTCAGAAAGAAAAGCCCTTATGTTAGC-3′; 3′-XhoI-Nub1L-UBA Domains 5′-CCGCTCGAGTTATCCTCCGTTGTGAGCAAGGG-3′. Expres-sion, purification, and ULP1 digestion were performed as previously described (Aichem et al, 2014, 2019). Expression and purification of GST and GST–FAT10 was also previously described (Schmidtke et al, 2006). The generation, expression, and purification of GST–NUB1L, GST–NUB1L–ΔUBA, and GST–NUB1L–ΔUBL was detailed before (Rani et al, 2012). The generation and expression of the N-terminal and C-terminal UBL domains of FAT10 has been described in the study by Aichem et al (2018).

### Generation of NUB1L–UBL–GFP and NUB1L–UBL–GFP–cytb

To express only the UBL domain of NUB1L as a fusion protein with GFP, or with a GFP and cytB fusion, we used pGEM-vectors provided by

Andreas Matouschek (Matouschek, 2000; Yu et al, 2016). We replaced ubiquitin in the vectors by the UBL domain of NUB1L by EcoRI and BamHI digestion after PCR amplification with the primers: NUB1L–UBL for: 5′-GTACATGAATTCATTAAAGAGGAGAAATTAACCATGATCGAGGTGTTTTTACCACC-3′ and NUB1L–UBL rev: 5′-ACTTGTCAACAGTACTTGAGAATTCCACGCCTTGTTCTTC-3′. Expression and purification was performed as described in the study by Yu et al (2016). For cloning of HA–NUB1L into yeast vector, p415GPD–Leu and pcDNA3.1–HA–NUB1L were digested with HindIII and XhoI, respectively. The insert was ligated into a HindIII- and SalI-digested p415GPD vector (kindly provided by Stefan Kreft, University of Konstanz, Germany). The correct insert was verified by sequencing. This vector allowed the yeast to grow on Leu-negative plates. The yeast strains SUB545 and the mutant SUB544, with a deletion of the nine amino acids residue tail (GSRRYDSRT) from the N-terminus of the α3-subunit, were constructed from SUB61 (MATα trp1-1 ura3-52 his3-Δ200 leu2-3,-113 lys2-801 [Finley et al, 1987], as described in the study by Groll et al [2000]). The strains were kindly provided by Suzanne Elsasser and Daniel Finley (Harvard University). Growth, transformation, selection, cycloheximide chase, lysis, and immunoprecipitation was performed exactly as described in the study by Rani et al (2012). The anti-PGK 1 Ab was purchased from Molecular Probes, order number A-6457. Anti-Flag IgG (order number F-1804) and anti-HA–IgG antibodies were purchased from Sigma-Aldrich (H9658).

### Generation of radioactive poly-ubiquitylated p53 and E6AP

For in vitro ubiquitylation experiments, E6-AP, UBE1, and UbcH7 were expressed in the baculovirus system or in *E. coli* BL21 as detailed elsewhere (Nuber et al, 1996). The vectors p53–pRcCMV and HA–E6AP–pcDNA were kindly provided by Martin Scheffner (University of Konstanz, Germany). The proteins were in vitro transcribed and translated in 50 μl aliquots as described by the supplier of the quick TNT-coupled transcription translation kit (Promega) with radioactive TranS 35 label methionine cysteine mixture (Hartmann Analytic). Unincorporated label was removed by three subsequent dilutions with assay buffer (25 mm Tris–HCl (pH 7.5), 50 mM NaCl, 1 mM dithiothreitol, and 4 mM MgCl$_2$) and concentrated with Amicon Ultra-0.5 filtration units. The reaction was incubated with 50 ng of UBE1, 50 ng of UbcH7, and 20 μg of ubiquitin (Sigma-Aldrich) in 120 μl with 2 mM ATP final concentration (Pelzer et al, 2007). Recombinant poly-ubiquitylated E6AP was generated as described in the study by Pelzer et al (2007) with the modification that UbcH5b was replaced by UbcH7.

### Generation of RPN10 and RPN1

The cDNAs for human GST- and/or hexa-His–tagged RPN10 and RPN1 were described before (Rani et al, 2012). The sequence of human RPN1 was mutated to introduce the T1 mutation of yeast D541A D548R E552R (Shi et al, 2016), (which is D541-A Q548-R E552-R in the human gene), with the primer: 5′-ATGATTCTGCTGTGG-GATGCGAAGGGTGGCCTCGCCGCGATTGACAAGTACCTGTACTCC-3′, and/or the T2 mutation L430A D431K Q434A Q435A (same sequence in yeast and human) with the primer: 5′-GTCCTGCAATGGAGCTG-TAACTTCCACTATCCTTCGGACCATCATGCGGAAGTCAGAGACTG-3′ and the quick change lightning kit according to the provider's protocol

(Agilent). Expression and purification of either GST- or hexa-His–tagged proteins was described before (Rani et al, 2012).

### Affinity measurement with the Octet instrument

Before the measurements, the sensors were hydrated in water for 10 min. All steps were carried out in 200 μl volume in 96 well plates. Before each measurement, the sensors were regenerated in 100 mM glycine buffer pH 3 for 30 s and washed in Octet buffer (PBS 0.1%, Triton X-100 1× ROTIBlock) for 30 s. This process was repeated three times. Nickel sensors were loaded with 20 mM NiCl for 60 s after the last regeneration step. GST served as negative control on the second sensor, when GST-tagged proteins were used, the second sensor was equilibrated in buffer when hexa-histidine–tagged proteins were used. The assay consisted of the steps baseline recording (in Octet buffer for 60 s), binding of 0.1 mg/ml GST, or hexa-histidine–tagged proteins to the sensor in 300 s, baseline 2 determination (Octet buffer for 60 s), association (varying concentrations of the ligand protein in Octet buffer for 300 s), dissociation (300 s in Octet buffer), and regeneration. All experiments were repeated at least three times with different protein preparations. For analysis, the curves were fitted, and dissociation constants were calculated using the 1:1 binding model (ForteBio Data Analysis 10.0 software). Recombinant human His6–NEDD8 was purchased from R&D systems (UL-813-500).

### Statistical analysis

Statistical analysis was performed in the programs Excel and GraphPad 6. The fluorescence intensity per min was calculated as mean between the time points 30–60 min. The error bars define the SD calculated by GraphPad. Calculation of the percentage was performed by this formula:

$$K = \text{single values of proteasome} + \text{substrate mean of proteasome} \times 100.$$

# Data Availability

This study includes no data deposited in external repositories. Supplementary data for this article are available.

# Supplementary Information

# Acknowledgements

We gratefully acknowledge Prof. Dr. Marcus Groettrup, who sadly passed away during the course of this study and we would like to dedicate this work to him, who devoted his life to the exploration of the immune system and the ubiquitin proteasome system. We thank Alfred L Goldberg for the GST–UBL and HIS–UIM plasmids for proteasome purification, and Daniel Finley and his group for providing us with the USP14-deficient MEF cell line and the α3ΔN

yeast strain. We are grateful for the contribution of plasmids by Andreas Matouschek. We acknowledge the group of Olga Mayans for help with Octet measurements. This study was supported by the German Research Foundation (DFG) SFB969 project C01 and C09 and by DFG grant GR 1517/25-1.

## Author Contributions

F Brockmann: formal analysis, investigation, and writing—original draft.

N Catone, C Wünsch, and F Offensperger: investigation.

M Scheffner: investigation and methodology.

G Schmidtke: conceptualization, supervision, investigation, and methodology.

A Aichem: supervision, investigation, and writing—review and editing.

## Conflict of Interest Statement

The authors declare that they have no conflict of interest.

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
