## [Reviewer comments · Life Science Alliance]

Life Science Alliance

FAT10 and NUB1L cooperate to activate the 26S proteasome

Florian Brockmann, Nicola Catone, Christine Wuensch, Fabian Offensperger, Martin Scheffner, Gunter Schmidtke, and Annette Aichem

DOI: <https://doi.org/10.26508/lsa.202201463>

Corresponding author(s): Annette Aichem, Biotechnology Institute Thurgau

Review Timeline:

Submission Date:	2022-03-23
Editorial Decision:	2022-04-28
Revision Received:	2023-02-20
Editorial Decision:	2023-03-27
Revision Received:	2023-04-26
Editorial Decision:	2023-05-02
Revision Received:	2023-05-02
Accepted:	2023-05-03

Scientific Editor: Novella Guidi

Transaction Report:

April 28, 2022

Re: Life Science Alliance manuscript #LSA-2022-01463-T

Prof. Marcus Groettrup
University of Konstanz
Department of Biology, Division of Immunology
Universitätsstrasse 10
Universitaetsstrasse 10
Konstanz 78457
GERMANY

Dear Dr. Groettrup,

Thank you for submitting your manuscript entitled "FAT10 and NUB1L cooperate to activate the 26S proteasome" to Life Science Alliance. The manuscript was assessed by expert reviewers, whose comments are appended to this letter. We invite you to submit a revised manuscript addressing the Reviewer comments.

Thank you for this interesting contribution to Life Science Alliance. We are looking forward to receiving your revised manuscript.

Sincerely,

B. MANUSCRIPT ORGANIZATION AND FORMATTING:

Reviewer #1 (Comments to the Authors (Required)):

Review of Brockmann et al.

The FAT-10-NUB1 system is an intriguing pathway for protein degradation in cells that is understudied. This well written study presents clear data on how FAT10 stimulates the peptidase activity of the 26S proteasome. These findings are certainly of mechanistic interest although mostly expected based on prior works showing a critical role of UBL-domains for enhancing peptide hydrolysis. However, the major weakness is that the analysis is incomplete because the authors seem unaware that activation of peptide hydrolysis is insufficient to catalyze protein degradation. The stimulation of degradation of a ubiquitin conjugate and thus unfolding of proteins also requires an unfolded substrate domain to activate the proteasome's ATPases. The authors should test this question, since several factors (e.g., PA28) can activate peptide hydrolysis, but are not known to do so for proteins. A number of other problems need to be addressed.

Major Problems

The citations (introductions and rationale) are out of date in multiple places. For example:
50-51- In describing Usp14s role (See recent papers of Finley's lab).. 51-52- and 58- onward. For allosteric inhibition by Usp14 and switch from inhibition to activation (see Cell review of Collins and Goldberg and papers of Peth (2010, 2011), Kim, and Collins. - The latter papers clearly indicate a second activation step, necessary for full activation (i.e. to increase in degradation of ubiquitinated proteins.

184-186- Doesn't this suggest that FAT10-NUB1L competes with Usp14 for sites on RPN1? Does FAT10 alone do so?

The supplementary data (EV2 A-C and 358-381) are not very convincing. The suggestion of possible substrate preference requires increasing concentrations of ubiquitinated and fatoylated proteins and comparison of affinity constants. Is this data available? How do their intracellular concentrations compare?

181 and 213-215- The authors should note that the activation by UBL is independent of USP14 and USP14 activates through its UBL domain (Kim, J. Biol Chem, PNAS).

The authors use gate opening (peptidase) assays to show an activation by FAT10 of 26S function. Their negative data regarding protein degradation in supplemental figure 2 is not convincing.

Figure EV 2A ... The 0 min points for Sic1, with and without FAT10+NUB1L, are very different. Why the discrepancy? It does not look like this assay is reliable.

Figure EV 2B ... Can authors quantify p53 labelling and graph the decay?

Figure EV 2C ... Why does the E6AP signal increase during the reaction with proteasomes and FAT10 + NUB1L? It does not look like this assay is reliable.

Minor Problems

Figure 1 B ... The x-axis should read NUB1L rather than NUB1.

Figure 1 F ... The legend explaining the color code is absent.

Text for Figure 2... They should emphasize that NUB1L and FAT10 are not endogenous to baker's yeast when explaining this experiment.

Figure 2 B ... Why does HA-NUB1L increase in the gate open mutant during the chase?

90- Height is the wrong word for migration in PAGE.

102-104 and 274-276 Figure 5 E... actually show a similar net increment in activity. The lower % increase in 5F is probably due to reaching the maximal limit in gate opening. Isn't it strange that ADP and ATP give similar increases?

106-117- Explore how FAT10 degradation is relevant to degradation of a FAT10-targeted substrate?

217- What does "excessive amounts" mean here?

254- The year 2009 is not "recently" but several papers are more recent and relevant here.

263- Is this surprising in light of their several UBL-domains?

338-339 Was not this the effect found for the UBL-domains by Collins?

Reviewer #2 (Comments to the Authors (Required)):

This is a comprehensive, well-designed and well-conducted study on the mechanism of activation of the 26S proteasome by FAT10 and its interaction partner NUB1L. The authors show that this activation is dependent on the binding of full length FAT10 to NUB1L and that this leads to the gate opening of the 20S proteasome. They further demonstrate that this activation is independent of ubiquitin and USP14. A model is proposed whereby inactive dimers of NUB1L are dissociated by competition of FAT10 for the UBA domains of NUB1L. This results in a free UBL domain of NUB1L which then binds to RPN1. A variety of experimental approaches were utilized. These findings significantly advance our understanding of the mechanism of a non-ubiquitin pathway of protein degradation by the 26S proteasome.

Some issues listed below should be addressed by the authors.

1. Fig. 1 The asterisks in this figure need to be defined in terms of the p values that they signify.

2. Fig. 1 E,F The concentrations of ADP, ATP and ATP S should be given in the figure legend.

3. Fig. 1H HA-FAT10cOc134L appears to have no effect on Z-GG-AMC hydrolysis as described in the text, yet the figure displays one asterisk when compared to the control. Explain.

4. Figs. 4B and EV1 Fig. 4B displays the affinity of NUB1L, FAT10 and NUB1L-bound GST-FAT10 binding to His6-RPN1. In Fig. EV1 in a similar experiment the binding affinity to RPN10 was studied. In this case NUB1L UBL was used rather than NUB1L. Why?

5. Fig. EV2 C and line 22 of text, It is stated on line 222 that "Degradation of a relatively long lived protein like poly-ubiquitylated E6AP did not show such a change in degradation over the time period of 60 min. Actually, the change observed is quite dramatic. Compare $t = 0$ to $t = 60$ min. for both and you will see that FAT10 + NUB1L does indeed slow the degradation of E6AP.

Typographical

Reference Choi et al., is incomplete.

Line 786 legend to Fig. EV1 His-RPN10 should be His6-RPN10.

Reviewer #3 (Comments to the Authors (Required)):

In this study, Brockmann and co-workers studied if the ubiquitin-like modifier FAT10 can stimulate proteasome activity by opening the entrance gate of the 26S proteasome. This is a relevant question as FAT10 similar to ubiquitin chains can target proteins for proteasomal degradation raising the question if similar mechanisms are in play in facilitating the degradation of FAT10-modified substrates. In a series of in vitro experiments, the authors provide data that support the model that FAT10 together with its binding partner NUB1L can open the entrance of the proteasome particle by a mechanism that is functionally similar to, but mechanistically distinct from, the USP14-dependent pore opening for degradation of ubiquitylated substrates. The molecular mechanism responsible for the FAT10-NUB1L-dependent stimulation of the proteasome remains unclear. While the in vitro data are strong, there is no validation that this mechanism stimulates degradation in the natural intracellular environment.

The authors raise an interesting question and provide in vitro experimental data that support the proposed model.

Major comments

The authors argue that this mechanism is important for stimulating the degradation of FAT10-modified proteins analogous to the Ubl-dependent opening for ubiquitylated proteins. The closest they get to showing this in vitro is by using FAT10ylated USE1, which they put forward as an example for a FAT10-modified substrate. However, USE1 is also the E2 that is responsible for FAT10 conjugation and as part of this process it will form a thiolester linkage between its active cysteine residue and the carboxy terminus of FAT10. This is essentially different from a substrate that is modified with FAT10 by an isopeptide linkage with FAT10. It is not clear from the description if the FAT10-activated USE1 with a thiolester linkage or FAT10ylated USE1 with an isopeptide bound. From the Materials & Methods it makes me suspect it is the former as the complex is run on a gel under non-reducing conditions. If so, it will be important to show the importance of the opening of the entrance with a truly FAT10ylated substrate.

As mentioned by the authors, a plausible explanation is that the catalytic sites are more easily accessible for the fluorogenic peptides because the proteasome will be open in the process of actively degrading FAT10. This is addressed by instead using the slowly degradable FAT10c0c134L mutant. It is concluded from this experiment that this is not the case as efficiently degraded wild-type FAT10 and slowly degraded FAT10c0c134L have the same effect. However, even though FAT10c0c134L is more stable, it is still degraded by the proteasome and the fact that this is a slow process does not mean that the gate will be less open during the degradation. I wonder if the authors can exclude this alternative interpretation based on this key experiment.

The importance of an unstructured initiation site is tested by including an unstructured C-terminus to FAT10. As I understood it, this experiment is based on a direct extrapolation of what has been found for ubiquitylated proteins. Do the authors have any data that support the importance of an unstructured initiation site for FAT10-mediated degradation? Have they tested whether FAT10-GFP-Cytb is faster degraded than FAT10-GFP? It is important to check this before conclusions are drawn about the importance, or lack thereof, of unstructured initiation sites for gate opening.

Fig 4A suggests that FAT10/NUB1L are competing with USP14 for binding to the 26S proteasome. Fig 5A suggests that they are not competing for binding to Rpn1. How do the authors explain this? This is an important issue because competition between USP14 and FAT10/NUB1L recruitment could provide an alternative explanation. Rpn1 is an integrated part of the 19S particle. How reliable are pulldown experiments performed with recombinant GST-Rpn1? Is the recombinant protein structurally correct in the absence of its binding partners?

Minor comments

In the result section is mentioned that Z-GGL-AMC is more efficiently cleaved in the native gel shown in Figure 1A whereas in the figure legend of the same figure Suc-LLVY-AMC is mentioned as the substrate.

Line 144 "Levy bodies" should be "Lewy bodies".

Fig 4C. Why is this experiment performed with FAT10 with the Cytb extension? It is not clear what NUB1L-UBL-GFP-Cytb is. I assume it is only the UBL domain of NUB1L. It may be better to have NUB1L in superscript to indicate that the full length protein is not part of the fusion.

Page 254. It is mentioned that recently USP14 was found to be an activator with a reference to a 2009 publication. Not that recent in my opinion and the link between USP14 and proteasome regulation indeed dates back from at least one decade.

Line 304-307. "In our experiments, we were able to show, that NUB1L formed dimers, and is binding to itself via its UBA- and UBL-domain, in vitro and that FAT10 competes with the UBL-domain of NUB1L for the binding of the three UBA domains of NUB1L, resolving the dimer/folding and therefore enabling the UBL-domain of NUB1L to bind to RPN1 (Fig 4C)." The part that is underlined is not shown but a model that is proposed by the authors.

Line 337. RPN23?? I guess that should be RAD23.

Point-by-point reply for Life Science Alliance manuscript #LSA-2022-01463-T with the title "FAT10 and NUB1L cooperate to activate the 26S proteasome" by Florian Brockman et al.

We are thankful to all reviewers for their constructive criticism and we hope that we now fulfill their requests. Line numbers have changed as compared to the original Word file. We have considered this in the point-by-point reply below. Moreover, we have added several text passages to the results and discussion section to make some points more clear.

Reviewer #1 (Comments to the Authors (Required)):

Review of Brockmann et al.

The FAT-10-NUB1 system is an intriguing pathway for protein degradation in cells that is understudied. This well written study presents clear data on how FAT10 stimulates the peptidase activity of the 26S proteasome. These findings are certainly of mechanistic interest although mostly expected based on prior works howing a critical role of UBL-domains for enhancing peptide hydrolysis. However, the major weakness is that the analysis s is incomplete because the authors seem unaware that activation of peptide hydrolysis is insufficient to catalyze protein degradation. The stimulation of degradation of a ubiquitin conjugate and thus unfolding of proteins also requires an unfolded substrate domain to activate the proteasomae's ATPases. The authors should test this question, since several factors (e.g., PA28 $\beta\alpha$) can activate peptide hydrolysis, but are not known to do so for proteins. A number of other problems need to be addressed.

Major Problems

The citations (introductions and rationale) are out of date in multiple places. For example: 50-51- In describing Usp14s role (See recent papers of Finley's lab).. 51-52- and 58- onward. tFor allosteric inhibition by Usp14 and switch from inhibition to activation (see Cell review of Collins and Goldberg and papers of Peth (2010, 2011), Kim, and Collins. - The latter papers clearly indicate a second activation step, necessary for full activation (i.e. to increase in degradation of ubiquitinated proteins.

We have changed the citations accordingly. Moreover, the introduction was also modified to explain

the given topic in more detail.

184-186- Doesn't this suggest that FAT10-NUB1L competes with Usp14 for sites on RPN1? Does FAT10 alone do so?

We also assumed that there might be a competition between FAT10-NUB1L and USP14 for RPN1 binding sites. Hence, we performed Octet measurements (lines 188 onward, or in revised manuscript lines 218 onward) and observed a better binding capacity of FAT10-NUB1L to RPN1 as compared to NUB1L. A competition of FAT10 with USP14 alone was not observed.

The supplementary data (EV2 A-C and 358-381) are not very convincing. The suggestion of possible substrate preference requires increasing concentrations of ubiquitinated and fatoylated proteins and comparison of affinity constants. Is this data available? How do their intracellular concentrations compare?

The requested data of the reviewer are not on hand. Although we searched for the intracellular concentrations of targeted proteins, we were not able to find them in literature. While suited in the extended figures, this topic still needs more investigation and serves here to strengthen the hypothesis of an ubiquitin-independent degradation pathway.

181 and 213-215- The authors should note that the activation by UBL is independent of USP14 and USP14 activates through its UBL domain (Kim, J. Biol Chem, PNAS).

We have modified our introduction accordingly. Furthermore, we have included a sentence for the binding of USP14 with its USP domain (new manuscript lane 218).

The authors use gate opening (peptidase) assays to show an activation by FAT10 of 26S function. Their negative data regarding protein degradation in supplemental figure 2 is not convincing.

As discussed in lane 364 and following, the degradation of poly-ubiquitylated proteins was slowed down by the addition of NUB1L and FAT10. Similar behavior of competition for the open gate proteasome was seen by the ubiquitin independent route of Ornithine decarboxylase (ODC) and its accelerator antizyme 1 (AZ1) and polyubiquitin (Zhang M et al, 2003). In these experiments, the initial binding step of poly-ubiquitin was able to inhibit the degradation of ODC *in vitro*. This result was interpreted as competition of these two proteins for one binding site. While this thesis works well for the competition of FAT10/NUB1L and poly-ubiquitin with its shuttle factors, Beenukumar et al. showed that ODC has specific receptors at the proteasome (Beenukumar RR et al, 2015). Regarding the high concentrations of FAT10 and NUB1L to 26S proteasome in comparison to “in vivo” data, we would not suggest to value the negative data regarding protein degradation in

supplemental figure EV2 as indicator for a non-activation of the 26S proteasome. It could be more of a selection for FAT10 degradation over poly-ubiquitylated proteins.

Figure EV 2A ... The 0 min points for Sic1, with and without FAT10+NUB1L, are very different. Why the discrepancy? It does not look like this assay is reliable.

The discrepancy come from the usage of different SIC-1 preparation. We believe that the experiment nevertheless is reliable since we compared the different time point internally to time point 0 either without or with FAT10 and NUB1L.

Figure EV 2B ... Can authors quantify p53 labelling and graph the decay?

We have not quantified the degradation rate of p53 because we think the stabilization of p53 in presence of FAT10 and NUB1L is very well visible.

Figure EV 2C ... Why does the E6AP signal _increase_ during the reaction with proteasomes and FAT10 + NUB1L? It does not look like this assay is reliable.

We agree with our reviewer. We have removed these data from Figure EV2 and from the text.

Minor Problems

Figure 1 B ... The x-axis should read NUB1L rather than NUB1.

We agree and have corrected the labelling.

Figure 1 F ... The legend explaining the color code is absent.

We apologize for the missing labeling. It is now included in Fig. 1F.

Text for Figure 2... They should emphasize that NUB1L and FAT10 are not endogenous to baker's yeast when explaining this experiment.

We have changed the text accordingly (revised manuscript, lanes 140-141): "This experiment was feasible as we had shown before that ectopically expressed FAT10 was degraded in yeast and that its degradation was accelerated by non-endogenous NUB1L to a similar extent as in mammalian cells".

Figure 2 B ... Why does HA-NUB1L_increase_ in the gate open mutant during the chase?

We think that the increase might derive from uneven blotting of HA-NUB1L. However, since we focused here on the degradation of FAT10, this should not matter.

90- Height is the wrong word for migration in PAGE.

We have changed the text accordingly (revised manuscript, lane 121).

102-104 and 274-276 Figure 5 E... actually show a similar net increment in activity. The lower % increase in 5F is probably due to reaching the maximal limit in gate opening. Isn't it strange that ADP and ATP give similar increases?

We assume that our reviewer refers to Fig. 1E and 1F instead of 5E and 5F. Indeed, the similar effect is interesting. However, Kim and colleagues (Kim HT & Goldberg AL, 2018) observed activation by the non-hydrolysable analog ATP γ S, the same as we did. We do not know why ADP was never used so far, but maybe be occupation of the ATP binding site with the lower affinity ADP is still sufficient.

106-117- Explore how FAT10 degradation is relevant to degradation of a FAT10-targeted substrate?

We apologize, but we are not sure if we understand the question correctly. Degradation of unconjugated FAT10 may prevent FAT10ylation of potential substrates, and save these proteins from degradation. So far we never saw differences in the degradation mode of FAT10, linear FAT10 fusion proteins or FAT10ylated substrates.

217- What does "excessive amounts" mean here?

An excessive amount describes here amounts, which are higher than stoichiometric amounts. We have mentioned this now in the text (revised manuscript, lane 254).

254- The year 2009 is not "recently" but several papers are more recent and relevant here.

We agree and have removed the word "recently" from the text.

263- Is this surprising in light of their several UBL-domains?

We wrote “surprisingly” because we were surprised that FAT10 can do this only together with NUB1L. Nevertheless, we agree with our reviewer and have removed this word from the text.

338-339 Was not this the effect found for the UBL-domains by Collins?

Collins and Goldberg (Collins GA & Goldberg AL, 2020) reported which UBL domain is able to activate the 26S proteasome. We do not see activation by the UBL domains alone, neither NUB1L nor any of the FAT10 domains alone are able to activate the 26S proteasome.

Reviewer #2 (Comments to the Authors (Required)):

This is a comprehensive, well-designed and well-conducted study on the mechanism of activation of the 26S proteasome by FAT10 and its interaction partner NUB1L. The authors show that this activation is dependent on the binding of full length FAT10 to NUB1L and that this leads to the gate opening of the 20S proteasome. They further demonstrate that this activation is independent of ubiquitin and USP14. A model is proposed whereby inactive dimers of NUB1L are dissociated by competition of FAT10 for the UBA domains of NUB1L. This results in a free UBL domain of NUB1L which then binds to RPN1. A variety of experimental approaches were utilized. These findings significantly advance our understanding of the mechanism of a non-ubiquitin pathway of protein degradation by the 26S proteasome.

Some issues listed below should be addressed by the authors.

1. Fig. 1 The asterisks in this figure need to be defined in terms of the p values that they signify.

An explanation of the asterisks and their respective p-value was added to the figure legend.

2. Fig. 1 E,F The concentrations of ADP, ATP and ATP γ S should be given in the figure legend.

The concentrations were added to the legend of Fig.1 E.

3. Fig. 1H HA-FAT10cOc134L appears to have no effect on Z-GG-AMC hydrolysis as described in the text, yet the figure displays one asterisk when compared to the control. Explain.

The Z-GGL-AMC hydrolysis of the proteasome incubated with HA-FAT10cOc134L appeared to be a little less but still was significant as compared to the control proteasome with an efficiency of 100%.

4. Figs. 4B and EV1 Fig. 4B displays the affinity of NUB1L, FAT10 and NUB1L-bound GST-FAT10 binding to His6-RPN1. In Fig. EV1 in a similar experiment the binding affinity to RPN10 was studied. In this case NUB1L Δ UBL was used rather than NUB1L. Why?

We wanted to investigate the change in the binding affinity of FAT10 to RPN10 when it is bound or unbound to NUB1L. A problem for this experiment is that the UBL-domain of NUB1L is able to bind to the VWA domain of RPN10, interfering with the measurement of FAT10 binding. Of course, the instrument cannot distinguish which of the proteins binds. Therefore, we only used the FAT10 binding UBA-domains of NUB1L because NUB1L Δ UBL still binds FAT10 but not RPN10.

5. Fig. EV2 C and line 22 of text, It is stated on line 222 that "Degradation of a relatively long lived protein like poly-ubiquitylated E6AP did not show such a change in degradation over the time period of 60 min. Actually, the change observed is quite dramatic. Compare $t = 0$ to $t = 60$ min. for both and you will see that FAT10 + NUB1L does indeed slow the degradation of E6AP.

After reading the comments of reviewer 1 and here of reviewer 2, we come to the conclusion that Fig. EV2C is indeed difficult to interpret. Therefore, we have decided to remove Fig. EV2C from the manuscript.

Typographical

Reference Choi et al., is incomplete.

Line 786 legend to Fig. EV1 His-RPN10 should be His6-RPN10.

We have now corrected this issues.

Reviewer #3 (Comments to the Authors (Required)):

In this study, Brockmann and co-workers studied if the ubiquitin-like modifier FAT10 can stimulate proteasome activity by opening the entrance gate of the 26S proteasome. This is a relevant question as FAT10 similar to ubiquitin chains can target proteins for proteasomal degradation raising the question if similar mechanisms are in play in facilitating the degradation of FAT10-modified substrates. In a series of in vitro experiments, the authors provide data that support the model that FAT10 together with its binding partner NUB1L can open the entrance of the proteasome particle by a mechanism that is functionally similar to, but mechanistically distinct from, the USP14-dependent

pore opening for degradation of ubiquitylated substrates. The molecular mechanism responsible for the FAT10-NUB1L-dependent stimulation of the proteasome remains unclear. While the in vitro data are strong, there is no validation that this mechanism stimulates degradation in the natural intracellular environment. The authors raise an interesting question and provide in vitro experimental data that support the proposed model.

Major comments

The authors argue that this mechanism is important for stimulating the degradation of FAT10-modified proteins analogous to the UbL-dependent opening for ubiquitylated proteins. The closest they get to showing this in vitro is by using FAT10ylated USE1, which they put forward as an example for a FAT10-modified substrate. However, USE1 is also the E2 that is responsible for FAT10 conjugation and as part of this process it will form a thiolester linkage between its active cysteine residue and the carboxy terminus of FAT10. This is essentially different from a substrate that is modified with FAT10 by an isopeptide linkage with FAT10. It is not clear from the description if the FAT10-activated USE1 with a thiolester linkage or FAT10ylated USE1 with an isopeptide bound. From the Materials & Methods it makes me suspect it is the former as the complex is run on a gel under non-reducing conditions. If so, it will be important to show the importance of the opening of the entrance with a truly FAT10ylated substrate.

We respectfully disagree with this comment. We have extensively shown that USE1 directly undergoes auto-FAT10ylation after being loaded with FAT10 and have even identified the lysine, which is FAT10ylated (Aichem A et al, 2014). Furthermore, the USE1-FAT10 conjugate is degraded by the proteasome and under *in vitro* conditions, the USE1-FAT10 conjugate appears to be quite stable and can not completely be dissociated under reducing conditions (Aichem A et al, 2014, Aichem A et al, 2010). Therefore, USE1 was published as the first identified FAT10 conjugation substrate. However, in the current study the main point was to show activation with a branched FAT10 substrate rather than by a linear fusion protein. Therefore, we used a branched, *in vitro* generated USE1-FAT10 conjugate. The nature of the bond is not relevant, as it does not matter how for instance poly-ubiquitin is connected, either by an isopeptide bound or by an artificial link via a sulphur bridge as shown in (Chen J et al, 2010, Jung JE et al, 2009). We have mentioned this now also in the main text.

As mentioned by the authors, a plausible explanation is that the catalytic sites are more easily accessible for the fluorogenic peptides because the proteasome will be open in the process of actively degrading FAT10. This is addressed by instead using the slowly degradable FAT10c0c134L mutant. It is concluded from this experiment that this is not the case as efficiently degraded wild-type FAT10 and slowly degraded FAT10c0c134L have the same effect. However, even though FAT10c0c134L is more

stable, it is still degraded by the proteasome and the fact that this is a slow process does not mean that the gate will be less open during the degradation. I wonder if the authors can exclude this alternative interpretation based on this key experiment.

We have used three different substrates, FAT10, FAT10c0c134L and FAT10-GFP. All three proteins have a different half-life and FAT10 and FAT10-GFP differ for example by a factor of approximately 2. However, together with NUB1L, all three proteins activate the proteasome to the same extent. Therefore, the speed of the degradation seems not to be so much important, but the binding to the proteasome. We have mentioned this now also in the main text.

The importance of an unstructured initiation site is tested by including an unstructured C-terminus to FAT10. As I understood it, this experiment is based on a direct extrapolation of what has been found for ubiquitylated proteins. Do the authors have any data that support the importance of an unstructured initiation site for FAT10-mediated degradation? Have they tested whether FAT10-GFP-Cytb is faster degraded than FAT10-GFP? It is important to check this before conclusions are drawn about the importance, or lack thereof, of unstructured initiation sites for gate opening.

We have shown in Aichem et al., (Aichem A et al, 2018) that FAT10 is degraded in a VCP-independent way by the proteasome. This degradation is dependent on the unstructured N-terminal MAPNASC sequence of FAT10, which, when removed, causes stabilization of FAT10. Concerning the unstructured region at the C-terminus of FAT10, this does not matter as FAT10-GFP is degraded, unlike RAD23 UBL-GFP. To be degraded Rad23 UBL-GFP needs the unstructured region of cytb at the end (Yu H et al, 2016). We were rather interested if grapping by the ATPases, which pull the unstructured region, has an influence on the activation. We mention this now in the text.

Fig 4A suggests that FAT10/NUB1L are competing with USP14 for binding to the 26S proteasome. Fig 5A suggests that they are not competing for binding to Rpn1. How do the authors explain this? This is an important issue because competition between USP14 and FAT10/NUB1L recruitment could provide an alternative explanation. Rpn1 is an integrated part of the 19S particle. How reliable are pulldown experiments performed with recombinant GST-Rpn1? Is the recombinant protein structurally correct in the absence of its binding partners?

We used recombinant GST-Rpn1 because this was also used by Shi et al., (Shi Y et al, 2016), who had identified the T1 and T2 binding sites in Rpn1 for Rad23 and USP14. Apparently, this protein behaves similar to Rpn1, which would be incorporated into the proteasome. In the same publication it was shown that the binding of USP14 to Rpn1 depends on the T2 site of Rpn1. Fig 4A shows displacement of USP14 from the 26S proteasome in the presence of FAT10 and NUB1 together. Fig 5A shows increased binding of USP14 under constant amounts of NUB1L with increased amounts of USP14.

The same figure shows that the binding of USP14 to RPN1 does not increase if FAT10 and NUB1L are present. It is apparently not as obvious as we expected, but quantification of ECL western blot signals usually gives unreliable results, and therefore, we did not do it.

Minor comments

In the result section is mentioned that Z-GGL-AMC is more efficiently cleaved in the native gel shown in Figure 1A whereas in the figure legend of the same figure Suc-LLVY-AMC is mentioned as the substrate.

We have corrected this mistake in the text. Fig. 1A shows degradation of Suc-LLVY-AMC (revised manuscript, lane 122).

Line 144 "Levy bodies" should be "Lewy bodies".

We have corrected this mistake.

Fig 4C. Why is this experiment performed with FAT10 with the Cytb extension? It is not clear what NUB1L-UBL-GFP-Cytb is. I assume it is only the UBL domain of NUB1L. It may be better to have NUB1L in superscript to indicate that the full length protein is not part of the fusion.

We agree and explain this now in more detail in the text: "On the left side FAT10 was incubated with GST-NUB1L and competed with increasing amounts of the UBL domain of NUB1L, fused to GFP-Cytb (named as NUB1L-UBL-GFP-Cytb)."

Page 254. It is mentioned that recently USP14 was found to be an activator with a reference to a 2009 publication. Not that recent in my opinion and the link between USP14 and proteasome regulation indeed dates back from at least one decade.

We agree and have changed this accordingly.

Line 304-307. "In our experiments, we were able to show, that NUB1L formed dimers, and is binding to itself via its UBA- and UBL-domain, in vitro and that FAT10 competes with the UBL-domain of NUB1L for the binding of the three UBA domains of NUB1L, resolving the dimer/folding and therefore

enabling the UBL-domain of NUB1L to bind to RPN1 (Fig 4C)." The part that is underlined is not shown but a model that is proposed by the authors.

Unfortunately, we do not see the underlined part. We nevertheless try to give an explanation. The direct dimerization is indeed not shown in Fig. 4C. However, we show the binding of the NUB1L UBL domain to NUB1L and that FAT10 dissolves this interaction. The increased binding capacity of NUB1L to the proteasome in presence of FAT10 is shown in Fig. 4A and 4B.

Line 337. RPN23?? I guess that should be RAD23.

Yes, RAD23 is correct.

References

- Aichem A, Anders S, Catone N, Rossler P, Stotz S, Berg A, Schwab R, Scheuermann S, Bialas J, Schutz-Stoffregen MC, et al (2018) The structure of the ubiquitin-like modifier fat10 reveals an alternative targeting mechanism for proteasomal degradation. *Nature communications* 9: 3321. doi:10.1038/s41467-018-05776-3
- Aichem A, Catone N, Groettrup M (2014) Investigations into the auto-fat10ylation of the bispecific e2 conjugating enzyme uba6-specific e2 enzyme 1. *The FEBS journal* 281: 1848-1859. doi:10.1111/febs.12745
- Aichem A, Pelzer C, Lukasiak S, Kalveram B, Sheppard PW, Rani N, Schmidtke G, Groettrup M (2010) Use1 is a bispecific conjugating enzyme for ubiquitin and fat10, which fat10ylates itself in cis. *Nature communications* 1:13: DOI:10.1038/ncomms1012.
- Beenukumar RR, Godderz D, Palanimurugan R, Dohmen RJ (2015) Polyamines directly promote antizyme-mediated degradation of ornithine decarboxylase by the proteasome. *Microbial cell* 2: 197-207. doi:10.15698/mic2015.06.206
- Chen J, Ai Y, Wang J, Haracska L, Zhuang Z (2010) Chemically ubiquitylated pcna as a probe for eukaryotic translesion DNA synthesis. *Nat Chem Biol* 6: 270-272. doi:10.1038/nchembio.316
- Collins GA, Goldberg AL (2020) Proteins containing ubiquitin-like (ubl) domains not only bind to 26s proteasomes but also induce their activation. *Proc Natl Acad Sci U S A* 117: 4664-4674. doi:10.1073/pnas.1915534117
- Jung JE, Wollscheid HP, Marquardt A, Manea M, Scheffner M, Przybylski M (2009) Functional ubiquitin conjugates with lysine-epsilon-amino-specific linkage by thioether ligation of cysteinyl-ubiquitin peptide building blocks. *Bioconjugate chemistry* 20: 1152-1162. doi:10.1021/bc800539p
- Kim HT, Goldberg AL (2018) Ubl domain of usp14 and other proteins stimulates proteasome activities and protein degradation in cells. *Proc Natl Acad Sci U S A* 115: E11642-E11650. doi:10.1073/pnas.1808731115
- Shi Y, Chen X, Elsasser S, Stocks BB, Tian G, Lee BH, Shi Y, Zhang N, de Poot SA, Tuebing F, et al (2016) Rpn1 provides adjacent receptor sites for substrate binding and deubiquitination by the proteasome. *Science* 351: doi:10.1126/science.aad9421
- Yu H, Kago G, Yellman CM, Matouschek A (2016) Ubiquitin-like domains can target to the proteasome but proteolysis requires a disordered region. *The EMBO journal* 35: 1522-1536. doi:10.15252/embj.201593147
- Zhang M, Pickart CM, Coffino P (2003) Determinants of proteasome recognition of ornithine decarboxylase, a ubiquitin-independent substrate. *The EMBO journal* 22: 1488-1496. doi:10.1093/emboj/cdg158

March 27, 2023

Re: Life Science Alliance manuscript #LSA-2022-01463-TR

Dr. Annette Aichem
Biotechnology Institute Thurgau
Unterseestrasse 47
Kreuzlingen 8280
Switzerland

Dear Dr. Aichem,

Thank you for submitting your revised manuscript entitled "FAT10 and NUB1L cooperate to activate the 26S proteasome" to Life Science Alliance. The manuscript has been seen by the original reviewers whose comments are appended below. While the reviewers continue to be overall positive about the work in terms of its suitability for Life Science Alliance, some important issues remain.

Our general policy is that papers are considered through only one revision cycle; however, given that the suggested changes are relatively minor, we are open to one additional short round of revision. Please note that I will expect to make a final decision without additional reviewer input upon resubmission.

Please submit the final revision within one month, along with a letter that includes a point by point response to the remaining reviewer comments.

To upload the revised version of your manuscript, please log in to your account: <https://lsa.msubmit.net/cgi-bin/main.plex>
You will be guided to complete the submission of your revised manuscript and to fill in all necessary information.

B. MANUSCRIPT ORGANIZATION AND FORMATTING:

Sincerely,

Reviewer #1 (Comments to the Authors (Required)):

We referees are both disappointed with the response put together by Brockmann et al., to our criticisms and those of Reviewer three. We still think that this manuscript contains some important observations and is near publication quality, but the authors

have made little or no effort to get there, and with a little effort, the conclusions would be more convincing for the readership. Further work addressing these points seems necessary. The authors' failure to address our concerns remains surprising.

Perhaps these deficiencies are due to the tragic loss of Prof. Groettrup and the inevitable chaos in the lives of these scientists, but we still expected more to be done in response to the critiques of the Referees..

We also found it unnecessarily challenging that in our version the line numbers never matched their reported lane (sic) numbers. A little effort would have been helpful in eliminating this ambiguity and problem for the readers.

For starters, no response was put forth (let alone experimental testing) of our chief complaint that because other factors (e.g., PA28 $\alpha\beta$) can activate peptide hydrolysis without clearly increasing protein degradation, the authors should examine other activities of proteasomes, such as ATP hydrolysis, and the role of an unfolded region in these activities.

Second, the authors did not resolve whether FAT10-NUB1L competes with Usp14 for binding to Rpn1. The Octet measurements don't test Usp14, and thus claiming no competition was observed is disingenuous. Perhaps in Figure 4A there might be information into this phenomenon, but (a) the gel is inappropriately cropped through the singly capped 26S particles. (b) Although Usp14 seems to decrease on doubly capped proteasomes, it seems to increase on the singly capped particles, and (c) it is not clear, since no relative molecular weights are provided, but NUB1 (I think they mean NUB1L) binds preferentially to the singly capped proteasomes. (d) All this is further complicated because we don't have a quality image of the proteasomes themselves to know if the ratio of 26S singly capped to doubly capped was affected by NUB1L or by FAT10 in these studies.

3rd Regarding the issues in the Supplemental figure S2, we may have been too subtle in our request for information.. In their final paragraph in the discussion (I have it as lines 423-425) the authors argue that the "competition on two poly-ubiquitin binding sites on the 26S proteasome leads to the assumption, that FAT10ylated proteins are favored for degradation." However, to make this claim the authors should ideally have the K_m and concentration information we asked about and certainly should measure the stability of FAT10-NUB1L under the conditions performed in supplemental figure 2. Absent of these vital bits of data, the authors conclusion is not resting on solid evidence.

4th Regarding the activation of proteasomes by Usp14's UBL domain, I see two issues. One, we don't see the change made that the authors said was done. This may be because my line numbers don't match theirs, but I don't see it in the place I would have expected it. Second, the authors very clearly got the model wrong in the abstract, suggesting that poly-ubiquitylated proteins displace "the inhibitory deubiquitylation enzyme USP14 from the 19S regulator subunit RPN1." That statement is at odds with Kuo and Goldberg (PNAS, 2017), and the recent work from Sakata and Finley's labs in Hung et al., (Nat Commun. 2022).

5th I still maintain that the data in supplemental figure 2 is unconvincing. The initial concentrations ubiquitylated Sic1 in the test condition vastly exceed the concentrations in the control condition. Thus, it is impossible to compare the rates of degradation, and no conclusions should be based on such data.

6th I don't understand the reluctance to improve the figure for p53 degradation by providing quantification. It's a simple aid to the reader to get as much information from that experiment as possible.

7th We understand that in Figure 2B the focus is on the degradation of HA-FAT10, nevertheless if the representative gel is demonstrating unusual blotting of something that can serve as a loading control, then the experiment should be repeated. It is puzzling that the nearby 100 kDa background band doesn't have this problem.

8th In respect to the response to reviewer 3's concern with USE1, it seems that the authors are over interpreting their data. The statement concerning " Surface SIINFEKLR results. The experiment in figure 3a reports that USE1 that has been allowed to auto-FAT10ylate, when combined with NUB1L, stimulates proteasome gate opening." In the caption the authors state that is FAT10ylated USE1 that increases proteasome activity. But from this experiment, can the authors show that all the FAT10 has been conjugated to USE1, and that there is neither USE1 charged with FAT10 or free FAT10 that could co-operate with NUB1L to lead to proteasome activity?

Other Minor points

- Abstract (page 2, line 29) - the language is imprecise about what is being modified by FAT10. It is clear to an expert, but other readers might wonder (based on context), is it isn't the proteasome or Usp14.
- Abstract (page 2, lines 32-33) - I don't understand what the authors mean by "exclusively in interplay."
- In Figure 1H the authors provide a mark of statistical significance for FAT10c0c134L. All other markers in that figure indicate activation. Here, the direction is presumable direction is inhibition. I recommend the authors mention that not only does FAT10c0c134L not stimulate peptidase activity, but it might also even inhibit it.
- In figure legend for Figure 1 the authors should clarify that NUB1L and FAT10 are added before gel electrophoresis, and they should define excess.
- In figure legend for Figure 1B, there is a missing hyphen between Z and GGL.
- In figure legend for Figure 1G, there is an inconsistency in nomenclature. GGL and LRR are peptides denominated in single letter abbreviations but Nle-Pro-Nle-Asp changes to the triple letter abbreviations.

- In figure legend for Figure 1H, FFAT10c0c134L should be offset by commas

Reviewer #3 (Comments to the Authors (Required)):

The authors have addressed my concerns. I have no further comments.

Point-by-point reply for manuscript #LSA-2022-01463-TR by Brockmann et al.

We are thankful to our reviewers for the valuable input to improve our manuscript. We regret that we could not satisfy our reviewers with the responses given in the first revision round, however, we hope that the second revision will now answer all remaining questions.

Reviewer #1 (Comments to the Authors (Required)):

We referees are both disappointed with the response put together by Brockmann et al., to our criticisms and those of Reviewer three. We still think that this manuscript contains some important observations and is near publication quality, but the authors have made little or no effort to get there, and with a little effort, the conclusions would be more convincing for the readership. Further work addressing these points seems necessary. The authors' failure to address our concerns remains surprising. Perhaps these deficiencies are due to the tragic loss of Prof. Groettrup and the inevitable chaos in the lives of these scientists, but we still expected more to be done in response to the critiques of the Referees.. We also found it unnecessarily challenging that in our version the line numbers never matched their reported line (sic) numbers. A little effort would have been helpful in eliminating this ambiguity and problem for the readers.

We again would like to apologize for the changes in the line numbers. As we have already explained in the first point-by-point reply, this issue emerged because we had to reconvert the pdf file from the first submission into a word file because we did not had access to the original files generated by Prof. Groettrup.

For starters, no response was put forth (let alone experimental testing) of our chief complaint that because other factors (e.g., PA28 $\alpha\beta$) can activate peptide hydrolysis without clearly increasing protein degradation, the authors should examine other activities of proteasomes, such as ATP hydrolysis, and the role of an unfolded region in these activities.

We reported some time ago that neither FAT10 nor NUB1L can bind to the 20S proteasome, neither alone nor together, what is different to PA28 $\alpha\beta$ which binds to the 20S complex (Schmidtke G et al, 2006). Binding of FAT10 and NUB1L requires the UBL domains of both proteins and requires subunits of the 26S complex (Rani N et al, 2012). The binding of PA28 $\alpha\beta$ is different from the binding of PA700, as the latter one induces a rotation in the α -subunits and displacement of a reverse turn loop that stabilizes the open-gate conformation (Rabl J et al, 2008). We could detect degradation of FAT10 alone or of a FAT10-DHFR fusion protein in presence of NUB1L *in vitro* by the 26S but not by the 20S proteasome (Schmidtke G et al, 2009). We did not measure ATP hydrolysis because binding of substrates was reported to enhance ATP hydrolysis. However, substrate-induced stimulation of ATPase activity did not appear to depend on, or be obligately linked to proteolysis because stimulation occurred similarly with proteasomes inhibited by MG132 or Epoxomicin (Kim YC et al, 2013). It was also reported that the PA700 complex can act like a chaperon, because unfolded and non-ubiquitylated proteins can stimulate the ATPase activity of PA700 (Braun BC et al, 1999). Even if we would report about the stimulation of the ATPase activity by FAT10 and NUB1L we would not be able to establish a link that this is required for function in degradation, nor could we exclude that the activation may be caused by some poorly folded contaminants in our preparation.

Concerning the role of an unfolded region, we have reported earlier that the FAT10 N-terminal part is unfolded and responsible for a VCP-independent degradation of FAT10 by the proteasome (Aichele et al, 2018). In addition, we would like to kindly point the attention of our reviewers to Fig. 3E. Here, we have investigated the impact of an unfolded region at the C-terminus of FAT10 on its ability to activate the proteasome. In detail, we have fused Cytb, described in Yu et al (Yu H et al, 2016) as an unstructured protein, to the C-terminus of FAT10. However, we did not see a difference in the ability of FAT10 or of FAT10-Cytb to activate the proteasome together with NUB1L.

Second, the authors did not resolve whether FAT10-NUB1L competes with Usp14 for binding to Rpn1. The Octet measurements don't test Usp14, and thus claiming no competition was observed is disingenuous. Perhaps in Figure 4A there might be information into this phenomenon, but (a) the gel is inappropriately cropped through the singly capped 26S particles.

We tried to measure USP14 affinity towards RPN1 with the Octet system, but did not measure any specific binding. Probably the His-tag on USP14 is responsible for this effect, as it was also binding to the control sensor. Moreover, we found a published affinity constant for USP14 together with UB-AMC. In one of the publications, which one of the referees suggested (Kuo CL & Goldberg AL, 2017) data were reported that the affinity of USP14 changes with and without polyubiquitin, and that USP14 dissociates rapidly without polyubiquitin. This may explain the poor binding in our Octet experiment, because this was done in the absence of polyubiquitin. We discuss these details now in lanes 378 - 385.

We did not attempt to claim a competition between USP14, NUB1L and FAT10. We investigated the possibility, of a competition, because USP14 was claimed to be essential. We report that USP14 is not required for proteasomal activation via FAT10 and NUB1L. We discuss these details also in lanes 378 - 385.

As recommended by our reviewers, we have changed the western blot in Fig. 4A and show now a larger part of the gel.

(b) Although Usp14 seems to decrease on doubly capped proteasomes, it seems to increase on the singly capped particles,

As now can be seen in the larger section of the western blot in Fig. 4A, USP14 does not increase on the single capped particles.

and (c) it is not clear, since no relative molecular weights are provided, but NUB1 (I think they mean NUB1L) binds preferentially to the singly capped proteasomes.

Yes, we mean NUB1L and we have corrected the labeling in the Western blot in Figure 4A.

Figure 4A shows a native gel. Molecular weight marker are not running correctly on native gels and thus cannot be used for correct labeling of protein sizes. However, as a substitute for the marker we have used the purified 26S proteasome in lane 1, labeled as "marker".

Concerning binding of NUB1L to single capped proteasomes, the impression might be correct, however, we do not know which might be the functional consequences of this observation.

(d) All this is further complicated because we don't have a quality image of the proteasomes themselves to know if the ratio of 26S singly capped to doubly capped was affected by NUB1L or by FAT10 in these studies.

We have shown in an earlier publication (Rani N et al, 2012) that FAT10 binds to Rpn10 and Nub1L to Rpn1 and Rpn10, which both belong to the regulatory particle of the 26S proteasome. We therefore suggest that it should not make a difference if the proteasome is single or double capped.

3rd Regarding the issues in the Supplemental figure S2, we may have been too subtle in our request for information.. In their final paragraph in the discussion (I have it as lines 423-425) the authors argue that the "competition on two poly-ubiquitin binding sites on the 26S sprteasome leads to the assumption, that FAT10ylated proteins are favored for degradation." However, to make this claim the authors should ideally have the Km and concentration information we asked about and certainly should measure the stability of FAT10-NUBL1 under the conditions performed in supplemental figure 2. Absent of these vital bits of data, the authors conclusion is not resting on solid evidence.

Unfortunately, we cannot show FAT10 degradation under the conditions shown in Fig S2. In Fig. S2 we measured degradation of ubiquitylated proteins over a time course of 20 min. FAT10 however, has a longer half-life of approximately 1 hour *in vitro* and *in cellulo* (Hipp MS et al, 2005, Schmidtke G et al, 2009, Schmidtke G et al, 2006). Moreover, under *in vitro* conditions, FAT10 degradation by the 26S proteasome is strictly dependent on the presence of NUB1L (Schmidtke G et al, 2009). Therefore, it is not possible to see a decline in the amount of FAT10 within the short time period of 20 minutes.

We apologize that we were not aware that Km and concentration information was requested. However, unfortunately, it is not possible to calculate the Km since it is not possible to measure the concentration of the radioactively labeled ubiquitylated proteins used in Fig. S2. Despite of this, we agree that we might have overinterpreted our findings and have now changed the sentences in lines 441-443 as follows:

"This gate-opening is ubiquitin and USP14 independent and our data might indicate a favored proteasomal degradation for FAT10ylated proteins since a decreased degradation of poly-ubiquitylated proteins was observed in the presence of FAT10 and NUB1L (Fig S2)."

4th Regarding the activation of proteasomes by Usp14's UBL domain, I see two issues. One, we don't see the change made that the authors said was done. This may because my line numbers don't match theirs, but I don't see it in the place I would have expected it.

We regret that our reviewer could not find the text change made in line 225:

"USP14 serves as a receptor protein for ubiquitylated proteins at the 19S regulator, leading to an open-gate state of the 26S proteasome (Peth A et al, 2009), either by binding via its UBL- or USP- domain (Aufderheide A et al, 2015, Kim HT & Goldberg AL, 2018) ."

Second, the authors very clearly got the model wrong in the abstract, suggesting that poly-ubiquitylated proteins displace "the inhibitory deubiquitylation enzyme USP14 form the 19S regulator subunit RPN1." That statement is at odds with Kuo and Goldberg (PNAS, 2017), and the recent work from Sakata and Finley's labs in Hung et al., (Nat Commun. 2022).

We apologize that we explained this in an incorrect way and have changed the sentence in the abstract as follows (lines 27-28):

"The interaction of the 19S regulatory particle of the 26S proteasome with ubiquitylated proteins leads to gate opening of the 20S core particle and increases its proteolytic activity by binding of the ubiquitin chain to the inhibitory deubiquitylation enzyme USP14 on the 19S regulator subunit RPN1."

In addition, we have now included the two publications suggested above by the reviewers in the introduction (lines 77-78)

5th I still maintain that the data in supplemental figure 2 is unconvincing. The initial concentrations ubiquitylated Sic1 in the test condition vastly exceed the concentrations in the control condition. Thus, it is impossible to compare the rates of degradation, and no conclusions should be based on such data.

We agree that the western blot shown in Figure S2A is difficult to interpret. We have now repeated the experiment and exchanged the western blot. In addition, we have also included a bar graph, showing the quantification of the ECL signals.

6th I don't understand the reluctance to improve the figure for p53 degradation by providing quantification. It's a simple aid to the reader to get as much information from that experiment as possible.

We agree and have now included a bar graph in Figure S2B, containing the quantification of the autoradiography signals.

7th We understand that in Figure 2B the focus is on the degradation of HA-FAT10, nevertheless if the representative gel is demonstrating unusual blotting of something that can serve as a loading control, then the experiment should be repeated. It is puzzling that the nearby 100 kDa background band doesn't have this problem.

We agree with our reviewers and have now repeated the experiment. In new Figure 2B we show now this new western blot, without such a blotting artefact.

8th In respect to the response to reviewer 3's concern with USE1, it seems that the authors are over interpreting their data. The statement concerning " Surface SIINFEKL results. The experiment in figure 3a reports that USE1 that has been allowed to auto-FAT10ylate, when combined with NUB1L, stimulates proteasome gate opening." In the caption the authors state that is FAT10ylated USE1 that increases proteasome activity. But from this experiment, can the authors show that all the FAT10 has been conjugated to USE1, and that there is neither USE1 charged with FAT10 or free FAT10 that could co-operate with NUB1L to lead to proteasome activity?

We can indeed show that we used only covalently modified USE1-FAT10 conjugate in our experiment, because the USE1-FAT10 conjugate was generated by a chemical reaction. To this aim, we expressed recombinant HA-tagged FAT10 C(0)C134L as a fusion protein with intein and a chitin binding domain. FAT10 C(0)C134L was subsequently purified via chitin beads. Cleavage from the column was done with Cystamin so that a modified FAT10 with a single SH group at the C-terminus was generated. A 10-fold molar excess of 5,5'- dithio-bis-(2-nitrobenzoic acid) (DTNB) was added to activate FAT10. Recombinant His6-tagged USE1 was then incubated with the activated FAT10 for 30, 60 or 120 minutes to perform the FAT10-USE1 ligation. The 6His-USE1-FAT10 conjugate was then purified by a Ni-NTA resin. As can be seen in the control western blots shown above, only USE1-FAT10 conjugate but no unconjugated USE1 was detectable (Fig. 1 A, lanes 4-6). Likewise, no unconjugated FAT10 was detected (Fig. 1B, lanes 3 and 4). Since in the chemical reaction no E1 activating enzyme for FAT10 (UBA6) was present, we can further exclude that FAT10-charged USE1 was present.

Figure 1

Other Minor points

- Abstract (page 2, line 29) - the language is imprecise about what is being modified by FAT10. It is clear to an expert, but other readers might wonder (based on context), is it isn't the proteasome or Usp14.

We agree with our reviewers and have changed the text as follows (line 29):

“Covalent modification of proteins with the cytokine inducible ubiquitin-like modifier FAT10 is an alternative signal for proteasomal degradation.”

- Abstract (page 2, lines 32-33) - I don't understand what the authors mean by "exclusively in interplay."

We have changed the text in the abstract as follows (lines 31-34):

“We also show that FAT10 is capable to activate all peptidolytic activities of the 26S proteasome, however only together with NUB1L, by binding to the UBA-domains of NUB1L and thereby interfering with NUB1L dimerization.”

- In Figure 1H the authors provide a mark of statistical significance for FAT10c0c134L. All other markers in that figure indicate activation. Here, the direction is presumable direction is inhibition. I recommend the authors mention that not only does FAT10c0c134L not stimulate peptidase activity, but it might also even inhibit it.

We completely agree with our reviewers and have changed the text in lines 154-155 as follows:

“FAT10c0c134L on its own was not able to activate the 26S proteasome and caused even a slight inhibition of the proteasome activity.”

- In figure legend for Figure 1 the authors should clarify that NUB1L and FAT10 are added before gel electrophoresis, and they should define excess.

We have changed the text as follows:

“(A) A native gel overlay assay was performed with human 26S proteasome in presence or absence of FAT10 and NUB1L. Samples were initially incubated for 15 min at 37°C with a 10-fold molar excess of NUB1L and FAT10 as compared to the 26S proteasome, before gel electrophoresis was performed.”

- In figure legend for Figure 1B, there is a missing hyphen between Z and GGL.

We have now corrected this mistake.

- In figure legend for Figure 1G, there is an inconsistency in nomenclature. GGL and LRR are peptides denominated in single letter abbreviations but Nle-Pro-Nle-Asp changes to the triple letter abbreviations.

We agree and have changed this into single letter code.

- In figure legend for Figure 1H, FFAT10c0c134L should be offset by commas

We agree and have changed this as follows: “The activity assay was performed as described in (B) using the native FAT10 as well as the stabilized form of FAT10, FAT10c0c134L, in the presence or absence of NUB1L.”

Reviewer #3 (Comments to the Authors (Required)):

The authors have addressed my concerns. I have no further comments.

We are happy that we could satisfy our reviewer and would like to thank him for his constructive criticism.

References:

Aichem A, Anders S, Catone N, Rossler P, Stotz S, Berg A, Schwab R, Scheuermann S, Bialas J, Schutz-Stoffregen MC, et al (2018) The structure of the ubiquitin-like modifier fat10 reveals an alternative targeting mechanism for proteasomal degradation. *Nature communications* 9: 3321. doi:10.1038/s41467-018-05776-3

- Aufderheide A, Beck F, Stengel F, Hartwig M, Schweitzer A, Pfeifer G, Goldberg AL, Sakata E, Baumeister W, Forster F (2015) Structural characterization of the interaction of ubp6 with the 26s proteasome. *Proc Natl Acad Sci U S A* 112: 8626-8631. doi:10.1073/pnas.1510449112
- Braun BC, Glickman M, Kraft R, Dahlmann B, Kloetzel PM, Finley D, Schmidt M (1999) The base of the proteasome regulatory particle exhibits chaperone-like activity. *Nature cell biology* 1: 221-226. doi:10.1038/12043
- Hipp MS, Kalveram B, Raasi S, Groettrup M, Schmidtke G (2005) Fat10, a ubiquitin-independent signal for proteasomal degradation. *Molecular and cellular biology* 25: 3483-3491.
- Kim HT, Goldberg AL (2018) Ubl domain of usp14 and other proteins stimulates proteasome activities and protein degradation in cells. *Proc Natl Acad Sci U S A* 115: E11642-E11650. doi:10.1073/pnas.1808731115
- Kim YC, Li X, Thompson D, DeMartino GN (2013) Atp binding by proteasomal atpases regulates cellular assembly and substrate-induced functions of the 26 s proteasome. *J Biol Chem* 288: 3334-3345. doi:10.1074/jbc.M112.424788
- Kuo CL, Goldberg AL (2017) Ubiquitinated proteins promote the association of proteasomes with the deubiquitinating enzyme usp14 and the ubiquitin ligase ube3c. *Proc Natl Acad Sci U S A* 114: E3404-E3413. doi:10.1073/pnas.1701734114
- Peth A, Besche HC, Goldberg AL (2009) Ubiquitinated proteins activate the proteasome by binding to usp14/ubp6, which causes 20s gate opening. *Molecular cell* 36: 794-804. doi:10.1016/j.molcel.2009.11.015
- Rabl J, Smith DM, Yu Y, Chang SC, Goldberg AL, Cheng Y (2008) Mechanism of gate opening in the 20s proteasome by the proteasomal atpases. *Molecular cell* 30: 360-368. doi:10.1016/j.molcel.2008.03.004
- Rani N, Aichele A, Schmidtke G, Kreft SG, Groettrup M (2012) Fat10 and nub1l bind to the vwa domain of rpn10 and rpn1 to enable proteasome-mediated proteolysis. *Nature communications* 3: 749.
- Schmidtke G, Kalveram B, Groettrup M (2009) Degradation of fat10 by the 26s proteasome is independent of ubiquitylation but relies on nub1l. *FEBS letters* 583: 591-594.
- Schmidtke G, Kalveram B, Weber E, Bochtler P, Lukasiak S, Hipp MS, Groettrup M (2006) The uba domains of nub1l are required for binding but not for accelerated degradation of the ubiquitin-like modifier fat10. *J Biol Chem* 281: 20045-20054.
- Yu H, Kago G, Yellman CM, Matouschek A (2016) Ubiquitin-like domains can target to the proteasome but proteolysis requires a disordered region. *The EMBO journal* 35: 1522-1536. doi:10.15252/emj.201593147

May 2, 2023

RE: Life Science Alliance Manuscript #LSA-2022-01463-TRR

Dr. Annette Aichem
Biotechnology Institute Thurgau
Unterseestrasse 47
Kreuzlingen 8280
Switzerland

Dear Dr. Aichem,

Thank you for submitting your revised manuscript entitled "FAT10 and NUB1L cooperate to activate the 26S proteasome". We would be happy to publish your paper in Life Science Alliance pending final revisions necessary to meet our formatting guidelines.

-please add the Twitter handle of your host institute/organization as well as your own or/and one of the authors in our system

Figure Check:

-please remove the Panel A from the S1 figure legend; since this is the only panel, we do not need it designated with a letter
-Figure 4A: please reduce the thickness of the thick black line down the middle of the blots

A. FINAL FILES:

B. MANUSCRIPT ORGANIZATION AND FORMATTING:

Sincerely,

May 3, 2023

RE: Life Science Alliance Manuscript #LSA-2022-01463-TRRR

Dr. Annette Aichem
Biotechnology Institute Thurgau
Unterseestrasse 47
Kreuzlingen 8280
Switzerland

Dear Dr. Aichem,

Thank you for submitting your Research Article entitled "FAT10 and NUB1L cooperate to activate the 26S proteasome". It is a pleasure to let you know that your manuscript is now accepted for publication in Life Science Alliance. Congratulations on this interesting work.

DISTRIBUTION OF MATERIALS:

Again, congratulations on a very nice paper. I hope you found the review process to be constructive and are pleased with how the manuscript was handled editorially. We look forward to future exciting submissions from your lab.

Sincerely,
